# CORRELATION-BASED SELF-SUPERVISION FOR FEW-SHOT TABULAR LEARNING

## ABSTRACT

Despite its paramount importance in many real-world applications, e.g., few-shot learning, self-supervision for tabular data remains challenging. The inherently heterogeneous nature of tabular data substantially impedes the generation of pseudo-labels with high fidelity. The commonly utilized random-based selection methodology largely ignores the complex relationships among features and might induce substantial noise into the self-supervision process, especially when the selected features have little relations with others. To address this issue, this paper proposes a simple yet effective solution: utilizing the correlation among original features to qualitatively evaluate the possible quality of pseudo-labels. Accordingly, a correlation-based randomness, rather than pre-defined uniform randomness, is proposed to select features for pseudo-label generation according to their overall correlations towards others. We employ our design in VIME, SCARF, STUNT and SAINT. The experimental results in various few-shot classification tasks reveal significant performance improvements across 50 OpenML datasets, compared to the original design. Code and datasets are available at the supplemental file.

## 1 INTRODUCTION

Tabular data is the most commonly used data form and is essential for numerous critical applications, e.g. medicine (Xu et al., 2019) and advertising (Luo et al., 2020). Generally, tabular data are trained on a specific task with a large set of labels. However, the acquisition of labeled samples can often be costly and sometimes impossible in many practical scenarios. Self-supervised learning, emerging as a learning paradigm that generates pseudo-labels, mitigates the dependence on manual labels and becomes increasingly popular for few-shot learning in which self-supervised models are fine-tuned with few labeled samples (Borisov et al., 2022). Few-shot learning has been extremely successful in certain domains, for example, computer vision (CV) and natural language processing (NLP) as pseudo-labels can be generated based on unique domain-specific properties in the field, e.g., color distortion (Zhang et al., 2016) and cropping (Chen et al., 2020) on the shapes invariants in CV, and token masking on token relations in NLP (Song et al., 2020).

In contrast to CV or NLP, tabular data is heterogeneous in nature and generally lacks explicit spatial or semantic structures(Wang et al., 2024). Thus, self-supervision for tabular data is rather challenging, as one needs to discover and exploit latent relations among features without domain knowledge support (Somepalli et al., 2021). Limited methods have been proposed for tabular data self-supervision and can generally be divided into four major categories (Yu et al., 2023): (1) Generative models: train to reconstruct the data from randomly corrupted data, e.g. VIME (Yoon et al., 2020) with multi-task autoencoder; (2) Contrastive learning-based models: learn invariance between the original and negative samples, e.g. SCARF (Bahri et al., 2021); (3) Meta-Learning: use K-Nearest Neighbor (KNN) to generate pseudo-labels, e.g., CATUS (Hsu et al., 2018) on embeddings and STUNT (Nam et al., 2023) on selected raw features. (4) Approaches mixed with more than one technique, e.g. SAINT (Ucar et al., 2021) a Transformer-based work that uses both reconstruction loss and InfoNCE contrastive loss. These self-supervised methods, albeit differences in forms, share a common design: using random-based methods for the generation of pseudo-labels or augmented views ( we use pseudo-labels to refer to both cases later) by treating different features equally.

However, the features in tabular data play unequal roles in self-supervision. Figure1a shows the correlation (Sedgwick, 2012) matrix for *diabetes* with various strengths of correlation. This difference would definitely influence the effects of self-supervision. Due to limited computational resources, we should give priority to features with high dependency towards others. According to information theory (Kullback, 1997), when pseudo-labels are independent of inputs, they essentially constitute noise with respect to inputs and result in low-quality models. Figure1b shows that the efficacy of self-supervision rapidly deteriorates when certain unrelated features

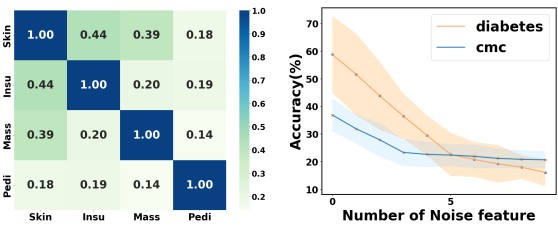

(a) Corr. matrix of diabetes     (b) Impacts of noises

Figure 1: Feature relations diversity and their impacts on self-supervision. (a) Feature pairs exhibit varying correlations (value calculated with Pearson's). (b) With the intrduction of additional unrelated features, the performance declines rapidly.

are added to *diabetes* and *cmc*. Two introduced unrelated features in self-supervision can result in a 20% accuracy loss for both datasets. It clearly shows that the features are not equally effective in self-supervision and we should not treat different features equally with randomness-based self-supervision.

In this paper, we go beyond the homogeneous assumption used in tabular learning and propose a simple yet effective solution: utilizing the correlation among original features to qualitatively evaluate and select the features for high-quality pseudo-label generation. The main contributions are:

**Beyond uniform random limitations:** We first reveal the limitations of existing self-supervision approaches in ignoring complex correlations among features in pseudo-label generation. We experimentally show that the uniform random selection during self-supervision might introduce a significant portion of noise into models.

**Correlation-based Randomness:** We propose a set of simple designs to integrate the relations among features to guide the pseudo-label generation process. We average the absolute values of the correlations of features to represent their overall correlation with other features: the greater the correlation, the higher the probability of the feature being selected. To mitigate the impacts from out-of-distribution correlations, a group-based selection strategy is proposed.

**Outstanding experimental results:** Our strategy is applied on four representative self-supervision baselines for tabular data and evaluated in 1-shot, few-shot, and multitask few-shot tasks across more than 50 OpenML datasets. Significant performance enhancements are observed in over 90% of the evaluated datasets for 1-shot learning scenarios, and approximately 80% for 5-shot learning contexts, substantiated by statistical confidence.

## 2 RELATED WORK

Self-supervision for tabular data can generally be categorized into three types of self-supervision tasks and their combinations.

**Generative models:** augment the data to train generative models and use the model to reconstruct from randomly corrupted data. Inspired by Masked Autoencoder (MAE)(Pathak et al., 2016), VIME (Yoon et al., 2020) and MAP (Lin et al., 2023) randomly corrupt the features to learn intrinsic feature relations from both the value and topology perspectives with a dual-task autoencoder. TabTransformer(Huang et al., 2020) uses a transformer instead of an autoencoder. SwitchTab(Wu et al., 2024) leverages an asymmetric encoder-decoder architecture on top of VIME. SubTab (Ucar et al., 2021) reconstructs complete features by strategically classifying them into subsets.

**Contrastive learning methods:** use contrastive learning frameworks for data augmentation. SCARF generates positive and negative samples through random data corruption and learns latent feature relations by controlling the distance between the positive and negative samples. TransTab (Wang & Sun, 2022) contextualizes columns and cells in tables with transformer encoders pre-trained with vertical-partition contrastive learning. PTaRL (Ye et al., 2023) constructs a prototype-based projection space and learns the disentangled representation around the global data prototypes.

**Meta-learning methods:** use clustering methods such as k-means to generate a pseudo-label for each unlabeled data and then train a model with meta-learning frameworks (Snell et al., 2017). CACTUS constructs pseudo-labels on a subset of intermediate representations, and UMTRA (Khodadadeh et al., 2019) uses N randomly sampled data to form an N-way 1-shot task. STUNT (Nam et al., 2023) extends CACTUS by randomly selecting subsets of raw features instead of using embeddings.

**Hybrid and pre-trained models:** Except for those works, many methods adopt more than one type of self-supervision tasks, e.g. SubTab (Ucar et al., 2021) with contrastive loss and distance loss and SAINT(Somepalli et al., 2021) performs attention over both rows and columns and pre-trains on a reconstruction loss and InfoNCE contrastive loss. Another stream of work adopts pre-trained language models for tabular learning, e.g., LIFT-ICL (Dinh et al., 2022) or P2T (Nam et al., 2024). They normally use the column meta-data to convert the records to sentences and limited work, e.g.,(Yan et al., 2024) transfers scalar values to discrete tokens. There are also works directly in training large models for tabular data, e.g., Xtab (Zhu et al., 2023). By treating different values as tokens, those works essentially use them uniformly.

# 3 METHOD

This section presents our design of feature correlations to support self-supervised processes.

## 3.1 NOTATION & PROBLEM DEFINITION

**Notation:** We have a labeled dataset $\mathbf{X}^l = \{\mathbf{x}^l_i, y^l_i\}^{N^l}_{i=1} \subseteq \mathbf{X} \times \mathbf{Y}$ the number of feature columns is $d$ with $n^l$ samples. We select an unlabeled dataset $X^u = \{\mathbf{x}^u_i\}^{N^u}_{i=1}$ and $\mathbf{X}^u \subseteq \mathbf{X}$ to generate pseudo-labels $Y_u$ accordingly. In few-shot settings, the cardinality of the labeled set is very small, e.g., one sample per class, while we have a sufficient amount of the unlabeled dataset, i.e., $N_u \gg N_l$. The symbol $\mathbf{c}_i$ is used to represent the i-th column and $\mathbf{x}_j$ denotes the j-th sample of $\mathbf{X}$.

**Problem Definition:** Formally, given a labeled and unlabeled dataset $\mathbf{X}^l$ and $\mathbf{X}^u$, $\mathbf{X}^l$ corresponding to a set of labels $\mathbf{Y}^l$, the goal for few-shot learning is to train a model $f : \mathbf{X}^l \rightarrow \mathbf{Y}^l$ parameterized by $f_\theta$, with the support of the self-supervised $h(\cdot)$ with parameter $h_\theta$ which is in turn supported by the transformation function t with parameter $t_\theta$. Thus, the learned model $h(\cdot)$ can minimize $\mathcal{L}$. The optimization problem can be expressed as :

$$\min_{f_\theta, h_\theta, t_\theta} \mathcal{L}\left(f(\mathbf{X}^l, \mathbf{Y}^l), h(\mathbf{X}^u, \mathbf{Y}^u), t(\mathbf{X}^u) \rightarrow \mathbf{Y}^u\right) \tag{1}$$

where $\mathcal{L}(\cdot)$ is the loss function of supervised optimization. Supervised works focus only on $f(\cdot)$, while most self-supervised work focuses on finding a good $h(\cdot)$ function, while largely ignoring the $t(\cdot)$ function. They generally adopt a uniform distribution in selecting inputs for label generation. Due to limited computational resources, we need to find an optimized selection function $t^*()$ to construct a set of effective pseudo-labels $\mathbf{Y}^{u*}$ for self-supervision.

## 3.2 PROPOSED METHOD

In this section, we introduce the design of our proposed correlation-based self-supervision.

### 3.2.1 OVERALL DESIGNS

To allow the model to learn the real latent structures among features, we need to select real informative features that have close relations with other features. In the machine learning domain, correlation coefficients have long been proposed to measure the strength of correlation between two continuous random variables. Figure 2 shows the overall process in the design and application of correlation-based self-supervision. This figure shows two major designs: 1) overall correlation calculation on how to represent the overall association strength of any particular feature; 2) group-based probability calculation on translating the diverse strengths into different feature selection possibilities for pseudo-label generation.

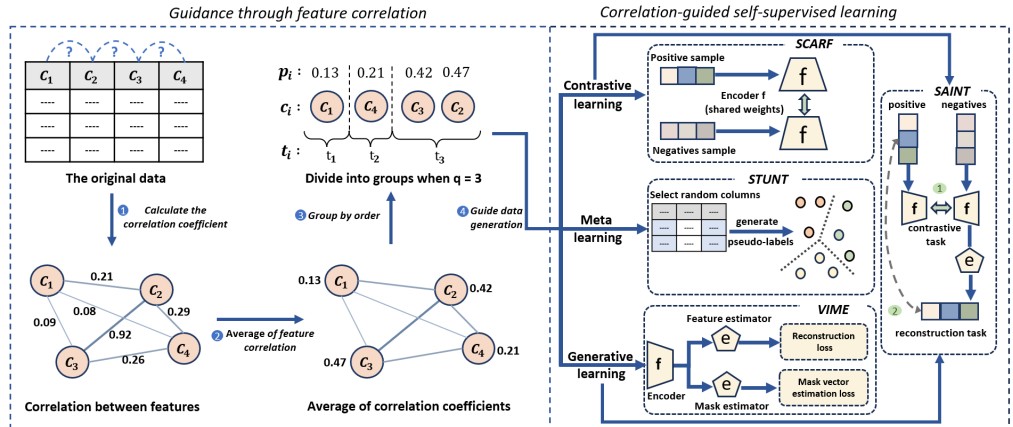

Figure 2: An overview of our correlation-based self-supervision for tabular learning with examplar integrations with diverse self-supervision methods, SAINT use both the contrative learning and the generative model

### 3.2.2 KEY PROCESSES

This subsection provides the key designs for the correlation-based selection probability.

**Averaged coefficients between features:** Due to limited training resources, it is important to employ the most informative feature for self-supervision. Furthermore, we can hardly get enough label signals to support this indication. Here, we choose to use the correlation coefficient for feature effectiveness evaluation. If a feature has high correlation towards other features, it is more likely to contain important information for other features which can be used for noise removal and/or strengthen salient relations during self-supervision. As shown in Figure 1, a feature's correlation towards other features might be highly unbalanced. Two highly correlated features might be totally unrelated to other features and are of little information towards self-supervision. Thus, rather than directly using paired feature correlations as SEFS(Lee et al., 2022), we choose to use the cumulative correlation coefficient to measure a feature's overall correlation. Then, for each column $c_i$, the averaged absolute values of the correlation coefficients between $c_i$ and other columns(e.g. $c_j$) are calculated by:

$$p_i = \sum_{j=0, j\neq i}^{j<d} |Cor(\mathbf{c}_i, \mathbf{c}_j)|/(d-1) \tag{2}$$

where $(Cor(\cdot))$ denotes correlation coefficients. There exist many different correlation metrics, e.g. Pearson and Spearman for different types of associations. $p_i$ denotes the averaged absolute correlation coefficients of $c_i$. Through Eq. 2, we obtain a vector $\mathbf{p} = [p_1, p_2, ..., p_d]$. Here, for categorical features, no specific embedding operation is applied to allow for a focused discussion on the effectiveness of our correlation design.

**Group-based feature probability calculation:** For previous self-supervised studies (SCARF, STUNT, and VIME), they randomly generated mask vectors $\mathbf{m} = [m_1, ..., m_d]^T \in \{0, 1\}^d$ to represent the selection or corruption of the features, with each feature being selected with a fixed probability. For our solution, it is natural to use the normalized $\mathbf{p}$ to generate the mask/selection probabilities. However, this design might be highly influenced by certain erroneous coefficients. It is highly possible that several very large/small $p_i$ might impact the overall distribution and the slight differences in features are easily obviated. It is especially true in heterogeneous tabular data as one correlation metric

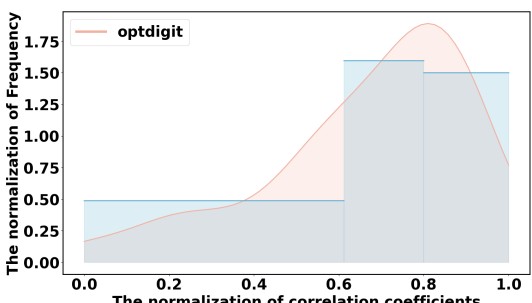

Figure 3: Group-based possibility calculation to mitigate impacts from erroneous correlations.

might not effectively handle different relations. Figure 3 shows the unbalanced correlation distribution in *optidigit* with the Pearson coefficient.

To mitigate those impacts, we introduce an equal frequency grouping strategy to divide all features into groups with equal numbers(3 groups in Figure 3). First, we sort the features in ascending order $\mathbf{p}_s$ from the column of features $\mathbf{p}$. Then, we divide the features equally into $q$ groups. The selection probabilities are assigned to each group rather than based on individual feature correlation coefficients. For a selection strategy with a minimal probability $r_{min}$ and a maximal probability $r_{max}$, we adopt the selection probability $g_i$ of group $i$ as follows

$$g_i = (r_{max} - r_{min}) * (i - 1)/q + r_{min}, i \in \{0...q-1\} \tag{3}$$

The selection possibility increases linearly with the order of the group. Here, $0 \leq r_{min} \leq r_{max} \leq 1$. When $r_{min} = r_{max}$, this strategy actually simplifies into a conventional selection strategy of equal probability. Then, the new selection vector $\mathbf{m^g}$ is generated, where $m_k$ is randomly sampled from a Bernoulli distribution with probability $g_j$ with feature $c_k$ belonging to the group $j^{th}$.

### 3.3 Specific Applications

**VIME:** leverages an autoencoder to predict the positions and original values of the corrupted data, thereby learning the intrinsic feature representation of the data. Here, we replace the $\mathbf{m}$ generated from random selection with our revised $\mathbf{m}$ based on group-based probability:

$$\tilde{\mathbf{x}} = \mathbf{m^g} \odot \bar{\mathbf{x}} + (1 - \mathbf{m^g}) \odot \mathbf{x} \tag{4}$$

The encoder $e$ and the pretext tasks (two estimators $s_m$ and $s_r$) are trained jointly as in Eq. 5.

$$\min_{e, s_m, s_r} \mathrm{E}_{\tilde{\mathbf{x}} \sim g_{\mathbf{m}}(\mathbf{x}, \mathbf{m})}[l_m(\mathbf{m^g}, \hat{\mathbf{m}}) + \alpha \cdot l_r(\mathbf{x}, \hat{\mathbf{x}})] \tag{5}$$

where $\hat{\mathbf{m}}$ for mask vector estimation and $\hat{\mathbf{x}}$ for feature vector estimation.

**STUNT:** is a meta-learning method that generates pseudo-labels by clustering randomly selected partial features. Here, we replace the $\mathbf{m}$ with our version $\mathbf{m^g}$. The selected features are then used for clustering to generate labels for model training according to Eq.6.

$$\min \frac{1}{N} \sum_{i=1}^{N} \min_{\tilde{y}_{u,i} \in \{0,1\}^k} \|\mathrm{sq}(\mathbf{x}_{u,i} \odot \mathbf{m^g}) - \mathrm{C}\tilde{y}_{u,i}\|_2^2 \tag{6}$$

where $\odot$ is the element-wise product and $\mathrm{sq}(\mathbf{x} \odot \mathbf{m^g})$ denotes a squeezing operation that removes the elements with a mask value of zero.

**SCARF:** utilizes a contrastive learning framework, generating positive and negative samples by randomly corrupting features. Instead of random corruption as in SCARF, we corrupt features proportionally to their correlation coefficients, as defined in Eq. 4. The corrupted data are then used to generate positive and negative samples.

**SAINT:** employs a Transformer with row- and column-wise attention and jointly optimizes reconstruction and contrastive objectives using CutMix and Mixup augmentations. We also follow Eq. 4 to corrupt features in the reconstruction view and the contrastive view generation process.

## 4 Experiments

In this section, we assess the effectiveness of our approach with a set of experiments.

### 4.1 Experimental Settings

**Datasets:** Similar to SCARF, we use 69 datasets from the public OpenML-CC18 benchmark (Vanschoren et al., 2014; Bischl et al., 2017) under the CC-BY license. As our solution relies on feature correlation, we remove datasets with fewer than five features. After this process, 50 datasets are selected: 35 datasets having only numerical features, 4 datasets having only categorical features, and 11 datasets having a mixture of numerical and categorical features. The categorical data are treated as

normal ordinal data, and we directly use correlation coefficients on those datasets without using any embedding technique. Detailed information on these data can be found in Appendix 3.

**Baselines:** To determine whether the introduced correlation-guided selection indeed facilitates enhanced performance, we applied this method to VIME, SCARF, SAINT, and STUNT. and compared them to the original versions. Five other strong baselines, including two meta-learning methods: CACTUS, UMTRA; one Transformer-based method TabTransformer (Huang et al., 2020); two strong supervised solutions: CatBoost (Prokhorenkova et al., 2018) and classical multi-layer perceptron (MLP) are also selected.

**Settings:** For all baselines, we strictly adhere to the configurations of the optimal hyperparameters as delineated in their papers. For SCARF and VIME, we use the uniform probability of 30% for masking. For our extension, we set group number $q$ to $\lfloor d/10 \rfloor + 1$. When dimension $d$ is greater than 100, $q$ equals 10 while $q = 2$ when $d$ is less than 10. Groups are assigned a linearly increasing probability with $r_1 = 0.2$, $r_2 = 0.4$ based on correlation order. We use the Adam optimizer (Kingma & Ba, 2014) with a learning rate $1e-3$, and weight decay $1e-4$. For the K-neighbor classifier, the number of neighbors is set to match the total number of categories in the dataset.

**Evaluation methods:** For all datasets, we selected 80% of the data for training and 20% of the data for testing. For STUNT with pseudo-label generation, 80% of the training data is used to generate pseudo-labels for model training, and 20% of the data as the validation set. Unless otherwise specified, LR is used as the classifier for the supervised learning and evaluation process, as it has the most stable results in few-shot scenarios. Results are obtained from 100 runs with 100 random seeds. Following STUNT, all models are trained for 10,000 epochs to achieve the stability and reliability of the experimental results in few-shot settings, see Appendix D.5. The discussion on hyperparameter settings can be seen in Appendix D.7.

## 4.2 DIFFERENT CORRELATION METRICS

In this section, we want to check which correlation metric is the most effective one. Eight metrics are evaluated, including: *Pearson* (Pearson, 1896), *Kendall* (Kendall, 1938), *Maximal Information Coefficient (MIC)*, *Distance Correlation(DC)* (Székely et al., 2007), *Chi-squared test(Chi)* (Pearson, 1900) and *mRMR* (Peng et al., 2005). Experiments are performed based on STUNT.

Table 1: Columns from left to right: Average accuracy, average relative improvement, average rank on 50 datasets, correlation calculation time (in seconds), and computational complexity.

| Method | Performance Comparison | | | Computation overhead & complexity | | | |
| | Avg Acc | ARI(%) | Avg Rank | karhunen | optdigit | diabetes | Complexity |
|---|---|---|---|---|---|---|---|
| STUNT | 54.76 | 0 | 7.56 | - | - | - | - |
| Spearman | 55.09 | 0.42 | 6.68 | 3.12 | 3.81 | 0.04 | $\mathcal{O}(m^2 n \log n)$ |
| Kendall | 54.92 | 0.04 | 6.96 | 2.42 | 2.89 | 0.02 | $\mathcal{O}(m^2 n \log n)$ |
| +Chi. | 55.96 | 2.14 | 4.9 | 4.92 | 13.64 | 0.03 | $\mathcal{O}(m^2 n)$ |
| MIC | 56.07 | 2.39 | 4.26 | 141.39 | 193.25 | 0.29 | $\mathcal{O}(m^2 n^2 \log n)$ |
| DC | 55.96 | 2.2 | 4.74 | 12.63 | 35.55 | 0.07 | $\mathcal{O}(m^2 n^2)$ |
| MI | 56.05 | 2.36 | 4.58 | 13.86 | 18.87 | 0.06 | $\mathcal{O}(m^2 n \log n)$ |
| mRMR | 56.11 | 2.40 | 3.96 | 31.57 | 93.13 | 0.13 | $\mathcal{O}(m^2 n \log n)$ |
| Pearson | **56.53** | **3.31** | **1.36** | **0.36** | **0.56** | **0.01** | $\mathcal{O}(m^2 n)$ |

Table 1 clearly shows that our solution obtains certain performance gains compared to the original version, with any correlation metric. Results are shown in Table 1. These results clearly support our design motivation that features with higher correlation might be more informative for self-supervision. Here, Pearson demonstrates a notable superiority over all other seven metrics, while mRMR ranks second. Spearman and Kendall are the ones with the worst performance. The reason contributing to its success lies in two folds. Firstly, as pointed out by (van den Heuvel & Zhan, 2022), Pearson coefficient can measure non-linear monotonic associations, in addition to linear relations. Secondly, those two relations are the most important and salient relations that are important for self-supervision, especially in few-shot learning with very limited label signals. In comparison, Spearman and Kendall exhibit poorer performance as they are more inclined towards handling rank-related relations which are hard to learn through self-supervision. Both MIC and Chi-square can identify linear and non-linear

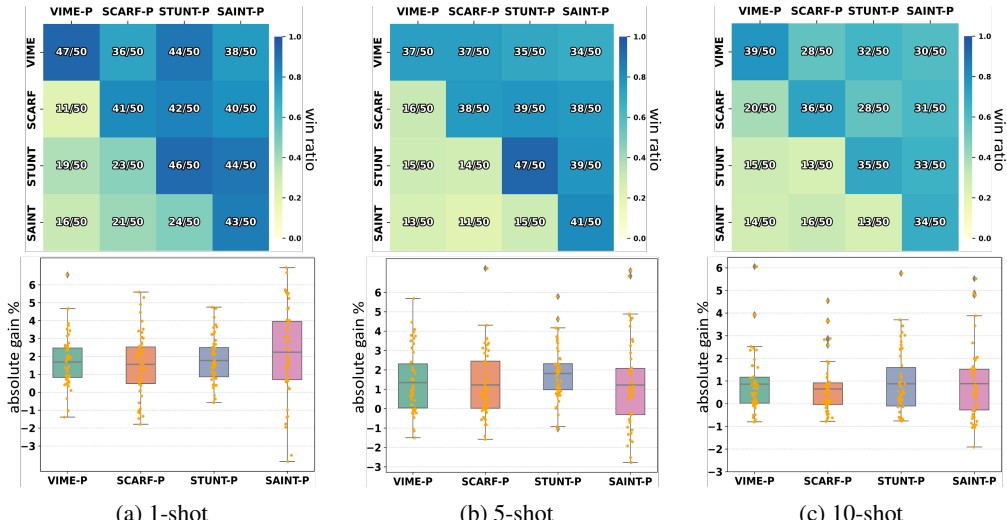

Figure 4: The performance comparison between four baselines and our enhanced versions. **Upper**: Win matrices with *-P v.s. original ones on 50 datasets with 100 runs, darker the better in the upper triangle. **Lower**: the absolute gain in accuracy over the original baselines.

relations and perform better than Spearman and Kendall. Their performance partially lies in the range differences: $[0, 1]$ for MIC and $[0, \infty)$ for Chi-squared test. A large value of correlations could significantly influence the overall cumulative correlation index of the Chi-squared test.

**Complexity Analysis:** Our solution might introduce some additional computational overhead for the correlation calculation. Table 1 shows their calculation time on different datasets and time complexity with different metrics. We can easily see that the Pearson is the most efficient in both execution time and complexity $O(m^2n)$, which is less than others, e.g. $O(m^2n^2)$ for Kendall and $O(m^2n^2 \log n)$ for MIC. Here, the execution times are collected via the Pandas.corr() function without further optimization. As cross-column correlations can be easily parallelized, we envisage much faster performance with GPU-based implementations (Chang et al., 2009). Thus, we think this overhead can be negligible as this operation can be precomputed offline.

## 4.3 FEW-SHOT CLASSIFICATION

Here, we report the results with our design on Pearson, denoted as *-P variants for different baselines.

**Overall improvements:** We perform performance comparisons in the 1, 5, 10-shot learning tasks on the 50 evaluated datasets. The upper parts of Figure 4 show the win matrices of our version and the original version for VIME, SCARF, SAINT, and STUNT in the 1, 5, 10 shot scenarios across 50 datasets. We can see that our variants achieve performance gain on more than 86% of datasets in 1 shot across different self-supervision baselines. The gains on more shots are gradually lower when the shot increases, as the classifier can already learn a better model with more labels. Those results clearly signify the effectiveness of using the correlation for pseudo-label generation. The lower parts of Figure 4 show absolute performance improvements. For one shot, VIME-P achieves more than 1% absolute performance improvement over VIME in 32 datasets, with 3.44% on average. Similarly, SCARF-P, SAINT-P, and STUNT-P are 31, 32, and 34, with average improvements of 2.70%, 2.09%, and 2.71%, respectively.

**Significance test:** The statistically significant tests are performed to validate the significance of improvements. In 1-shot, our variants achieve $P \leq 0.01$ in the second quartile(Q2). Even in Q3, $P$ is still much less than 0.05. For instance, VIME-P in 1-shot, 45 out of 47 results have $P <0.05$. Even for the few inferior results, P is normally large. For Q2, all variants have P >0.05. The minimal of VIME-P in 1-shot is P=0.07. Thus, our results have clear advantages over the original versions with statistical significance. Table B.3 in the appendix shows the detailed results with similar conclusions.

**Results on selected datasets:** Table 2 compares our approaches with existing supervised and self-supervised baselines in 1 and 5 shots. Generally, methods with self-supervision achieve better results

Table 2: Accuracy(%) on 10 datasets from the OpenML-CC18 benchmark averaged with 100 random seeds with LR classfier. ARI means Average Relative Improvements(%).

| Method / Dataset | karhunen | optdigit | cmc | diabetes | semeion | dna | splice | texture | first | steel |
|---|---|---|---|---|---|---|---|---|---|---|
| | | | | | # shot 1 | | | | | |
| CatBoost | 55.15 | 61.56 | 38.75 | 55.28 | 41.98 | 43.60 | 40.46 | 56.45 | 17.33 | 34.11 |
| MLP | 52.62 | 62.39 | 38.10 | 56.49 | 41.44 | 40.11 | 39.29 | 55.55 | 17.65 | 32.28 |
| LR | 58.41 | 67.37 | 38.41 | 51.18 | 45.32 | 43.82 | 37.69 | 53.84 | 16.43 | 34.57 |
| KNN | 57.63 | 66.89 | 37.92 | 52.84 | 42.46 | 42.97 | 38.84 | 50.36 | 17.12 | 33.77 |
| CACTUS | 55.79 | 64.10 | 36.73 | 56.62 | 43.75 | 56.41 | 40.36 | 61.04 | 17.31 | 33.39 |
| UMTRA | 46.01 | 41.24 | 36.47 | 55.87 | 30.93 | 30.21 | 37.47 | 56.06 | 16.33 | 26.48 |
| TabTransformer | 52.87 | 62.81 | 41.08 | 54.22 | 42.68 | 39.96 | 41.32 | 54.96 | 17.39 | 31.57 |
| VIME | 62.93 | 75.35 | 35.04 | 56.15 | 51.74 | 54.37 | 42.21 | 55.43 | 16.11 | 34.28 |
| SCARF | 57.92 | 69.86 | 35.07 | 55.29 | 52.76 | 51.67 | 38.68 | 54.06 | 18.28 | 34.53 |
| SAINT | 59.46 | 75.52 | 34.12 | 51.57 | 47.82 | 49.9 | 42.84 | 53.58 | 11.94 | 31.67 |
| STUNT | 71.06 | 75.15 | 36.87 | 58.79 | 56.41 | 67.31 | 41.67 | 62.38 | 18.49 | 34.68 |
| VIME-P | 66.16 | 76.79 | 39.71 | 58.90 | 58.29 | 57.51 | 42.31 | 57.30 | 18.44 | 36.91 |
| SCARF-P | 63.51 | 71.09 | 38.49 | 57.27 | 54.08 | 54.44 | 42.65 | 59.35 | 19.03 | 36.63 |
| SAINT-P | 65.03 | 73.75 | 36.17 | 54.36 | 54.78 | 55.62 | 43.54 | 57.19 | 15.83 | 33.09 |
| STUNT-P | 73.58 | 79.83 | 39.01 | 60.32 | 59.66 | 71.49 | 43.90 | 63.89 | 20.61 | 37.91 |
| ARI(%) | 6.92 | 1.89 | 8.72 | 4.12 | 8.87 | 7.20 | 4.37 | 5.58 | 15.65 | 6.89 |
| | | | | | # shot 5 | | | | | |
| CatBoost | 80.75 | 82.19 | 43.58 | 62.02 | 63.46 | 62.85 | 41.18 | 70.12 | 19.77 | 40.17 |
| MLP | 82.73 | 80.63 | 41.11 | 64.96 | 64.57 | 58.56 | 39.55 | 69.49 | 19.56 | 41.56 |
| LR | 80.61 | 81.13 | 40.61 | 61.29 | 62.89 | 60.95 | 40.03 | 67.57 | 19.50 | 40.76 |
| KNN | 80.60 | 80.73 | 40.85 | 61.57 | 63.09 | 59.97 | 38.27 | 69.36 | 19.45 | 37.53 |
| CACTUS | 81.17 | 81.37 | 38.42 | 65.47 | 63.43 | 79.34 | 39.29 | 68.97 | 20.32 | 40.04 |
| UMTRA | 76.23 | 70.61 | 37.56 | 60.42 | 40.18 | 34.59 | 38.04 | 60.27 | 19.81 | 33.52 |
| TabTransformer | 81.15 | 81.72 | 41.23 | 59.47 | 61.56 | 61.58 | 39.74 | 68.47 | 16.89 | 38.48 |
| VIME | 82.63 | 82.40 | 39.26 | 62.43 | 64.04 | 68.50 | 41.73 | 70.61 | 19.56 | 41.05 |
| SCARF | 81.78 | 82.34 | 39.25 | 64.34 | 62.24 | 69.06 | 44.93 | 70.27 | 19.13 | 42.51 |
| SAINT | 82.02 | 83.12 | 37.66 | 60.32 | 64.39 | 69.29 | 39.34 | 68.48 | 20.25 | 40.29 |
| STUNT | 84.65 | 87.49 | 42.47 | 67.16 | 73.48 | 80.14 | 42.72 | 73.10 | 20.05 | 45.11 |
| VIME-P | 84.74 | 83.92 | 41.22 | 65.34 | 68.13 | 70.88 | 45.67 | 74.11 | 21.52 | 42.24 |
| SCARF-P | 84.65 | 83.90 | 40.52 | 65.70 | 69.48 | 72.68 | 47.25 | 73.56 | 22.53 | 42.98 |
| STUNT-P | 86.95 | 91.58 | 42.85 | 66.23 | 74.91 | 83.52 | 46.22 | 73.79 | 24.67 | 46.02 |
| SAINT-P | 83.16 | 83.75 | 38.38 | 64.57 | 65.42 | 68.34 | 46.18 | 73.15 | 19.55 | 39.74 |
| ARI(%) | 2.54 | 2.29 | 2.76 | 3.11 | 5.39 | 2.89 | 10.05 | 4.35 | 11.84 | 1.16 |

than pure state-of-the-art supervised ones, e.g., CatBoost. VIME-P and other variants adapted with our strategy enjoy further performance improvements, from 1.89% $\sim$ 15.65% in 1-shot and 1.16% $\sim$ 11.84% in 5-shots. In contrast to models employing completely random selection or feature corruption, our method strategically selects or corrupts more features with higher correlation. This deliberate approach enables the model to assimilate more pertinent information, leading to a substantial increase in classification accuracy.

## 4.4 MULTI-TASK LEARNING

In this section, we evaluate our method in few-shot multi-task learning. We use the emotions dataset from OpenML consisting of a variety of audio data with 72 numerical features and multiple binary labels: amazed-surprised, happy-please, relaxing-calm, quiet-still, sad-lonely, and angry-aggressive. Due to page limits, we here only illustrate major results and details can be found in Appendix C. In the 1-shot classification, all three variants achieve significant improvements. VIME-P achieved average relative performance improvements of 4.42% while SCARF-P with 6.40% and STUNT-P with 5.87%. In the 5-shot task, VIME-P achieves an averaged relative improvement of 4.05%, while SCARF-P with 3.12% and STUNT-P with 3.36%. Our variants also outperform other strong baselines. The results clearly show the importance of selecting the appropriate features for self-supervision.

## 4.5 ANALYSIS EXPERIMENTS

In this section, we perform a set of experiments to verify the correctness of our basic designs.

**Ranges of possibilities:** Different self-supervision baselines usually have one basic parameter, the possibility of selecting a feature for label generation or view generation. As our solution introduces a possibility range(*Variance*) towards the original one fixed rate(*Mean*), we performed the experiments with the STUNT model in a 1-shot classification task to check the influence of both *Mean* and

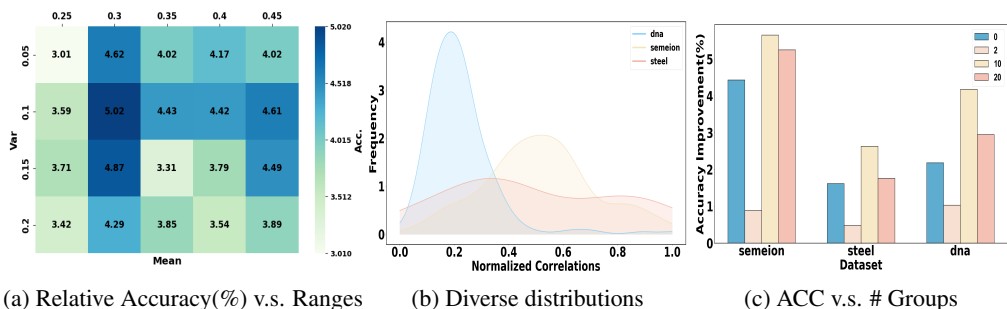

(a) Relative Accuracy(%) v.s. Ranges  (b) Diverse distributions  (c) ACC v.s. # Groups

Figure 5: (a) Average accuracy improvements(%) of STUNT-P with various ranges between Mean± Var. (b) The diverse distributions of Pearson coefficients on three datasets. (c) The accuracy of STUNT-P v.s # groups, 0 denotes feature-wise probability.

*Variance*. Figure 5a shows performance improvements with respect to STUNT( *Mean* = 0.3). We can easily observe that when the *mean* is 0.3, self-supervision usually has good results. This finding matches the default settings of many self-supervision baselines, e.g., VIME, SCARF which set possibility 0.3. When the *variance* is set to 0.1 with a possible range of [0.2, 0.4], the best results can be observed. When *variance* is too large, e.g., *variance* is 0.2, the most informative features might have a 50% chance to be selected. The remaining features might contain enough information for effective self-supervision. In comparison, while the *variance* is too small, the chance of selecting unrelated features would increase, which will definitely deteriorate the performance.

**Number of groups:** We also studied the impacts of the number of groups towards different groups. Here, 0 means one feature one bin. Figure 5b shows the normalized correlation distributions of three datasets. Each has its own distribution characteristics. Figure 5c shows the impacts of the number of groups. For all datasets, *10 groups* usually achieves the best results. When the distribution is even, for example, in the steel dataset, the difference between *10 groups* and *0 groups* is narrow. Here, if the distribution concentrates in certain regions, the advantages of grouping are very obvious. Of course, if we have too many groups, e.g. *20 groups*, its performance is close to *0 group*, loss of the tolerance to unbalanced distributions. In contrast, too few groups, e.g. *2* make too abrupt a difference between features. Only a slightly improved performance than the original version can be observed.

**Additional analysis:** To verify the robustness of our solution, we test its performance on synthetic datasets with linear, non-linear, and non-monotonic relations, real-world datasets with weak correlations, different types of noises, and unbalanced samples. We observe consistent and significant improvements on all those datasets(Appendix E). We also point out that the performance would significantly deteriorate if we select features with reverse correlation order(Appendix D.3).

**LLM-based or enhanced methods:** We also compared our solution with LLM-based models, e.g. LIFT-ICL and P2T, and explored the possibility of applying ChatGPT instead of correlations to see whether LLM can better guide the selection process than correlation(Appendix D.4). From our tentative experiments, we found that the performance of large models in learning tabular data with few samples cannot yet reach the performance of self-supervision based on correlation-based self-supervision enhancement. We think this can be an interesting area for further investigation.

## 5 CONCLUSION

In this paper, we study, for the first time, the challenge of selecting appropriate features for self-supervised tabular data learning. Due to the heterogeneous nature of tabular data, albeit its significant importance, this problem has been largely underexplored. To address this gap, we introduce the novel application of the different correlation coefficients to measure the correlation among feature columns in tabular data. Our primary concept involves selecting a larger proportion of feature columns exhibiting higher correlations with all other columns, while incorporating a smaller proportion of columns with lower correlations. This approach enables the model to learn more information between feature columns through correlation. We substantiate the effectiveness of our method through rigorous testing in few-shot experimental settings. We anticipate that our contributions pave the way for more fruitful and intriguing directions in the realm of tabular data learning.

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

# A EXPERIMENTAL DETAILS

## A.1 USE OF LLMS

In this paper, we include LLM-based tabular learning methods as baseline models for a more comprehensive comparison, given the recent progress of large language models in this area. Details can be found in Appendix D.4. Additionally, we also used LLMs to help with language polishing throughout the manuscript.

## A.2 DATASETS

50 datasets are selected from the public OpenML-CC18 benchmark: 35 datasets having only numerical features, 4 datasets having only categorical features, and 11 datasets having a mixture of numerical and categorical features. Table 3 shows the detailed information of these datasets.

Table 3: The description of attributes from 50 datasets selected from OpenML.

| Dataset Id | Class | Shape | Feature | Nume. | Cate. | Name |
|---|---|---|---|---|---|---|
| 16 | 10 | 2000*64 | 64 | 64 | 0 | mfeat-karhunen |
| 28 | 10 | 5620*64 | 64 | 64 | 0 | optdigit |
| 23 | 3 | 1473*9 | 9 | 2 | 7 | cmc |
| 37 | 2 | 738*8 | 8 | 8 | 0 | diabetes |
| 1501 | 10 | 1593*256 | 256 | 256 | 0 | semeion |
| 40670 | 3 | 3186*180 | 180 | 0 | 180 | dna |
| 46 | 3 | 3190*60 | 60 | 0 | 60 | splice |
| 458 | 4 | 841*70 | 70 | 70 | 0 | analcatdata |
| 40499 | 11 | 5500*40 | 40 | 40 | 0 | texture |
| 1475 | 6 | 6118*51 | 51 | 51 | 0 | first |
| 40982 | 7 | 1941*27 | 27 | 27 | 0 | steel |
| 40979 | 10 | 2000*240 | 240 | 240 | 0 | mfeat-pixel |
| 14 | 10 | 2000*76 | 76 | 76 | 0 | mfeat-fourier |
| 12 | 10 | 2000*216 | 216 | 216 | 0 | mfeat-factors |
| 22 | 10 | 2000*47 | 47 | 47 | 0 | mfeat-zernike |
| 1485 | 2 | 2600*500 | 500 | 500 | 0 | madelon |
| 1487 | 2 | 2534*72 | 72 | 72 | 0 | ozone-level-8hr |
| 44 | 2 | 4601*57 | 57 | 57 | 0 | spambase |
| 1494 | 2 | 1055*41 | 41 | 41 | 0 | qsar-biodeg |
| 1510 | 2 | 1055*41 | 30 | 30 | 0 | wdbc |
| 6332 | 2 | 540*37 | 37 | 18 | 19 | cylinder-bands |
| 4538 | 5 | 9873*32 | 32 | 32 | 0 | GesturePhaseSegmentationProcessed |
| 4534 | 2 | 11055*30 | 30 | 0 | 30 | PhishingWebsites |
| 4134 | 2 | 3751*1776 | 1776 | 1776 | 0 | Bioresponse |
| 40994 | 2 | 540*18 | 18 | 18 | 0 | climate-model-simulation-crashes |
| 40984 | 7 | 2310*16 | 16 | 16 | 0 | segment |
| 40978 | 2 | 3279*1558 | 1558 | 3 | 1555 | Internet-Advertisements |
| 40966 | 8 | 1080*77 | 77 | 77 | 0 | MiceProtein |
| 40701 | 2 | 5000*20 | 20 | 16 | 4 | churn |
| 3 | 2 | 3196*36 | 36 | 0 | 36 | kr-vs-kp |
| 38 | 2 | 3772*29 | 29 | 6 | 23 | sick |
| 32 | 10 | 10992*16 | 16 | 16 | 0 | pendigits |
| 31 | 2 | 1000*20 | 20 | 7 | 13 | credit-g |
| 307 | 11 | 990*12 | 12 | 10 | 2 | vowel |
| 300 | 26 | 7797*617 | 617 | 617 | 0 | isolet |
| 29 | 2 | 690*15 | 15 | 6 | 9 | credit-approval |
| 23381 | 2 | 500*12 | 12 | 1 | 11 | dresses-sales |
| 188 | 5 | 736*19 | 19 | 14 | 5 | eucalyptus |
| 182 | 6 | 6430*36 | 36 | 36 | 0 | satimage |
| 1497 | 4 | 5456*24 | 24 | 24 | 0 | wall-robot-navigation |
| 1480 | 2 | 583*10 | 10 | 9 | 1 | ilpd |
| 1478 | 6 | 10299*561 | 561 | 561 | 0 | har |
| 1068 | 2 | 1109*21 | 21 | 21 | 0 | pc1 |
| 1067 | 2 | 2109*21 | 21 | 21 | 0 | kc1 |
| 1063 | 2 | 522*21 | 21 | 21 | 0 | kc2 |
| 1053 | 2 | 10885*21 | 21 | 21 | 0 | jm1 |
| 1050 | 2 | 1563*37 | 37 | 37 | 0 | pc3 |
| 1049 | 2 | 1458*37 | 37 | 37 | 0 | pc4 |
| 54 | 4 | 846*18 | 18 | 18 | 0 | vehicle |
| 6 | 26 | 20000*16 | 16 | 16 | 0 | letter |

## A.3 SETTINGS & BASELINES:

**Settings:** In addition to the experiments mentioned in Figure 5a, for other experiments referenced in the main text, the hyperparameters are set as follows: $r_1 = 0.2$, $r_2 = 0.4$, and the learning rate of the

Adam optimizer (Kingma & Ba, 2014) is set to $1e - 3$. The MLP uses the ReLU activation function. Regarding the model's other hyperparameters, we strictly adhere to the configurations of the optimal hyperparameters as delineated in their respective original papers. We use Adam optimizer (Kingma & Ba, 2014) with learning rate $1e - 3$, and weight decay $1e - 4$. The logistic regression model is specified with 10k iterations. As for the K-neighbors classifier, the number of neighbors is set to match the total number of categories in the dataset.

**Baselines:** *CatBoost* (Prokhorenkova et al., 2018) is one of the gradient boosted decision tree algorithms. CatBoost consecutively builds decision trees in a way that reduces loss compared to previous trees. In configuring CatBoost hyperparameters for our study, we set iterations to 10k as it reaches best performance in the few-shot settings, max depth to 8, learning rate to 0.001, bagging temperature to 0.6, L2 leaf regularization to 3, and leaf estimation iterations to 1, with careful consideration to balance model complexity and optimize predictive performance.

*VIME* is a self-supervised learning method which extracts useful representations by corrupting random features and then predicting the corrupted location. For VIME pre-training, we follow the best hyperparameters suggested from the original paper. Using VIME representations, we perform k-nearest neighbor classification and logistic regression.

*SCARF* uses the framework of contrastive learning to randomly corrupt data and generate positive and negative samples. Making the model better learn the features of the data by controlling the distance between positive and negative samples. For SCARF pre-training, we follow the best hyperparameters suggested from the original paper. Using SCARF representations, we perform k-nearest neighbor classification and logistic regression.

*CACTUS* is an unsupervised meta-learning method which runs a clustering algorithm on a representation trained with self-supervised learning in order to self-generate the tasks. We follow the hyperparameters suggested in the original paper.

*STUNT* is an unsupervised meta-learning method whose task is constructed by randomly selecting features and using k-means to generate pseudo-labels.

*SAINT* is a Transformer-based model tailored for tabular learning. It captures both row-wise and column-wise dependencies through interleaved attention mechanisms, and employs masked token modeling for self-supervised pre-training.

For hyperparameters, some of them are shown here and the other part follows the advice in the original paper.

## B  LOW-SHOT AND FEW-SHOT CLASSIFICATION

In this section, we will present the test results for the remaining 40 datasets in 1-shot classification and 5-shot classification, as well as the test data for 50 datasets in 10-shot classification.

### B.1  FEW-SHOT CLASSIFICATION

From Table 4, Table 5, and Table 6, we observe the standard deviations for the 10 datasets mentioned in the main text, along with experimental data for the remaining 40 datasets. It can be observed that there is a significant improvement in the performance of the model after applying our proposed method.

For instance, in the 1-shot classification task, compared to the original version of STUNT, the application of our method in STUNT-P resulted in a notable increase in accuracy from 95.89% to 99.61% on analcatdata dataset from OpenML, representing an improvement of nearly 4 percentage points. Concurrently, the standard deviation decreased from 6.25 to 4.13. In the 5-shot classification task, VIME-P, after applying our method to the original VIME, exhibited an improvement in accuracy from 68.80% to 70.34% and a reduction in standard deviation from 5.63 to 3.99.

### B.2  LOW-SHOT CLASSIFICATION

In our experimental evaluation with an increased number of available labels, specifically in the 10-shot classification, we conducted thorough experiments to validate the efficacy of our proposed

Table 4: 1-shot test accuracy(%) and standard deviations on 50 datasets from the OpenML-CC18 benchmark, averaged with 100 random seeds.

| | # shot = 1 | | | | | | | | | | | |
|---|---|---|---|---|---|---|---|---|---|---|---|---|
| Id | STUNT | STUNT-P | VIME | VIME-P | SCARF | SCARF-P | SAINT | SAINT-P | LR | KNN | CACTUS | UMTRA |
| 16 | 71.06$_{\pm5.86}$ | 73.58$_{\pm5.82}$ | 62.93$_{\pm4.1}$ | 66.16$_{\pm3.65}$ | 57.92$_{\pm3.59}$ | 63.51$_{\pm3.73}$ | 59.46$_{\pm4.43}$ | 65.03$_{\pm3.72}$ | 55.15$_{\pm3.45}$ | 52.62$_{\pm2.96}$ | 58.41$_{\pm3.85}$ | 55.79$_{\pm7.66}$ |
| 28 | 75.15$_{\pm5.79}$ | 79.83$_{\pm5.29}$ | 75.35$_{\pm2.67}$ | 76.79$_{\pm2.75}$ | 69.86$_{\pm2.77}$ | 71.09$_{\pm2.94}$ | 75.52$_{\pm3.09}$ | 73.75$_{\pm2.64}$ | 61.56$_{\pm3.03}$ | 62.39$_{\pm3.24}$ | 67.37$_{\pm3.39}$ | 64.1$_{\pm6.98}$ |
| 23 | 36.87$_{\pm5.89}$ | 39.01$_{\pm5.76}$ | 35.04$_{\pm5.29}$ | 39.71$_{\pm4.14}$ | 35.07$_{\pm4.18}$ | 38.49$_{\pm5.37}$ | 34.12$_{\pm5.19}$ | 36.17$_{\pm3.81}$ | 38.75$_{\pm3.66}$ | 38.1$_{\pm3.17}$ | 38.41$_{\pm3.3}$ | 36.73$_{\pm8.27}$ |
| 37 | 58.79$_{\pm13.93}$ | 60.32$_{\pm13.34}$ | 56.15$_{\pm8}$ | 58.9$_{\pm10.2}$ | 55.29$_{\pm11.63}$ | 57.27$_{\pm8.98}$ | 51.57$_{\pm8.47}$ | 54.36$_{\pm9.9}$ | 55.28$_{\pm11.16}$ | 56.49$_{\pm11.89}$ | 51.18$_{\pm11.15}$ | 56.62$_{\pm16.07}$ |
| 1501 | 56.41$_{\pm7.06}$ | 59.66$_{\pm5.86}$ | 51.74$_{\pm3.66}$ | 58.29$_{\pm3.7}$ | 52.76$_{\pm3.15}$ | 54.08$_{\pm3.9}$ | 47.82$_{\pm4.14}$ | 54.78$_{\pm3.89}$ | 41.98$_{\pm4.94}$ | 41.44$_{\pm4.49}$ | 45.32$_{\pm4.08}$ | 43.75$_{\pm8.36}$ |
| 40670 | 67.31$_{\pm11.51}$ | 71.49$_{\pm11.34}$ | 54.37$_{\pm6.17}$ | 57.51$_{\pm5.81}$ | 51.67$_{\pm7.1}$ | 54.44$_{\pm6.97}$ | 49.9$_{\pm6.19}$ | 55.62$_{\pm6.22}$ | 43.6$_{\pm8.57}$ | 40.11$_{\pm8.66}$ | 43.82$_{\pm8.63}$ | 56.41$_{\pm14.2}$ |
| 46 | 41.67$_{\pm8.35}$ | 43.9$_{\pm8.08}$ | 42.21$_{\pm4.73}$ | 42.31$_{\pm4.32}$ | 38.68$_{\pm4.38}$ | 42.65$_{\pm4.47}$ | 42.84$_{\pm4.24}$ | 43.54$_{\pm3.99}$ | 40.46$_{\pm5.6}$ | 39.29$_{\pm5.82}$ | 37.69$_{\pm5.79}$ | 40.36$_{\pm8.77}$ |
| 40499 | 62.38$_{\pm6.25}$ | 63.89$_{\pm4.13}$ | 55.43$_{\pm2.57}$ | 57.3$_{\pm2.86}$ | 54.06$_{\pm7.3}$ | 59.35$_{\pm3.91}$ | 53.58$_{\pm2.37}$ | 57.19$_{\pm2.92}$ | 56.45$_{\pm2.39}$ | 55.55$_{\pm2.75}$ | 53.84$_{\pm2.64}$ | 61.04$_{\pm5.38}$ |
| 1475 | 18.49$_{\pm4.76}$ | 20.61$_{\pm4.73}$ | 16.11$_{\pm2.7}$ | 18.44$_{\pm2.57}$ | 18.28$_{\pm2.52}$ | 19.03$_{\pm2.23}$ | 11.94$_{\pm2.22}$ | 15.83$_{\pm2.75}$ | 17.33$_{\pm1.84}$ | 17.65$_{\pm1.88}$ | 16.43$_{\pm1.56}$ | 17.31$_{\pm6.86}$ |
| 40982 | 34.68$_{\pm6.46}$ | 37.91$_{\pm6.14}$ | 34.28$_{\pm4.38}$ | 36.91$_{\pm3.75}$ | 34.53$_{\pm6.53}$ | 36.63$_{\pm4.75}$ | 31.67$_{\pm3.96}$ | 33.09$_{\pm3.87}$ | 34.11$_{\pm3.94}$ | 32.28$_{\pm3.97}$ | 34.57$_{\pm4.04}$ | 33.39$_{\pm8.99}$ |
| 458 | 93.16$_{\pm6.25}$ | 96.93$_{\pm4.13}$ | 90.91$_{\pm2.57}$ | 91.84$_{\pm2.86}$ | 87.11$_{\pm7.3}$ | 88.71$_{\pm3.91}$ | 88.47$_{\pm2.32}$ | 89.17$_{\pm2.51}$ | 84.05$_{\pm3.58}$ | 85.32$_{\pm3.93}$ | 82.58$_{\pm3.49}$ | 75.82$_{\pm6.39}$ |
| 40979 | 79.76$_{\pm6.93}$ | 80.47$_{\pm6.6}$ | 63.65$_{\pm2.75}$ | 67.09$_{\pm2.61}$ | 65.98$_{\pm3.71}$ | 68.39$_{\pm3.31}$ | 60.78$_{\pm3.2}$ | 64.26$_{\pm2.89}$ | 76.51$_{\pm4.13}$ | 75.77$_{\pm4.68}$ | 78.58$_{\pm4.78}$ | 77.75$_{\pm7.85}$ |
| 14 | 54.21$_{\pm6.07}$ | 55.04$_{\pm6.76}$ | 52.42$_{\pm3.68}$ | 53.66$_{\pm3.24}$ | 50.87$_{\pm4.06}$ | 52.98$_{\pm2.49}$ | 53.19$_{\pm4.13}$ | 53.77$_{\pm3.36}$ | 51.13$_{\pm3.4}$ | 52.26$_{\pm3.36}$ | 51.51$_{\pm3.89}$ | 50.8$_{\pm8.26}$ |
| 12 | 76.19$_{\pm6.53}$ | 78.64$_{\pm6.25}$ | 74.98$_{\pm2.43}$ | 75.84$_{\pm2.38}$ | 70.23$_{\pm4.77}$ | 75.13$_{\pm2.64}$ | 75.6$_{\pm2.23}$ | 76.46$_{\pm2.15}$ | 74.77$_{\pm4.22}$ | 72.78$_{\pm4.46}$ | 73.56$_{\pm3.99}$ | 73.25$_{\pm9.26}$ |
| 22 | 51.43$_{\pm5.82}$ | 51.83$_{\pm6.2}$ | 46.12$_{\pm3.38}$ | 48.14$_{\pm4.09}$ | 45.11$_{\pm4.25}$ | 48.63$_{\pm3.32}$ | 47.76$_{\pm3.46}$ | 49.46$_{\pm4.55}$ | 49.51$_{\pm3.19}$ | 47.83$_{\pm3.74}$ | 48.26$_{\pm3.32}$ | 46.71$_{\pm5.78}$ |
| 1485 | 51.88$_{\pm5.06}$ | 52.81$_{\pm5.61}$ | 50.63$_{\pm2.45}$ | 51.45$_{\pm2.45}$ | 50.78$_{\pm3.97}$ | 51.61$_{\pm3.99}$ | 46.58$_{\pm2.48}$ | 50.62$_{\pm2.38}$ | 50.52$_{\pm2.88}$ | 49.47$_{\pm2.65}$ | 49.93$_{\pm2.72}$ | 49.48$_{\pm7.23}$ |
| 1487 | 56.72$_{\pm19.34}$ | 57.73$_{\pm18.87}$ | 51.33$_{\pm11.19}$ | 54.86$_{\pm10.92}$ | 55.26$_{\pm14.11}$ | 56.31$_{\pm7.82}$ | 49.07$_{\pm11.53}$ | 52.18$_{\pm10.79}$ | 54.2$_{\pm16.99}$ | 54.95$_{\pm16.99}$ | 53.17$_{\pm16.97}$ | 54.6$_{\pm22.29}$ |
| 44 | 78.26$_{\pm13.6}$ | 79.32$_{\pm16.19}$ | 73.32$_{\pm11.82}$ | 75.67$_{\pm9.05}$ | 70.5$_{\pm10.91}$ | 72.89$_{\pm9.99}$ | 69.38$_{\pm12.31}$ | 74.09$_{\pm8.84}$ | 75.99$_{\pm11.23}$ | 76.04$_{\pm11.56}$ | 74.57$_{\pm11.41}$ | 75.03$_{\pm14.26}$ |
| 1494 | 60.54$_{\pm11.26}$ | 62.25$_{\pm12.99}$ | 55.41$_{\pm5.78}$ | 56.76$_{\pm10.28}$ | 58.88$_{\pm10.4}$ | 59.33$_{\pm7.88}$ | 51.38$_{\pm6.13}$ | 54.85$_{\pm10.71}$ | 56.9$_{\pm8.58}$ | 56.74$_{\pm8.36}$ | 57.02$_{\pm8.91}$ | 56.73$_{\pm11.32}$ |
| 1510 | 88.51$_{\pm4.38}$ | 91.18$_{\pm3.98}$ | 85.99$_{\pm2.35}$ | 88.58$_{\pm2.09}$ | 86.94$_{\pm7.45}$ | 90.28$_{\pm1.51}$ | 82.76$_{\pm2.48}$ | 85.09$_{\pm2.07}$ | 85.26$_{\pm1.57}$ | 86.2$_{\pm2.35}$ | 85.88$_{\pm1.96}$ | 85.06$_{\pm4.86}$ |
| 6332 | 55.7$_{\pm2.76}$ | 57.36$_{\pm3.15}$ | 54.4$_{\pm3.68}$ | 55.7$_{\pm1.92}$ | 55.12$_{\pm3.49}$ | 57.23$_{\pm3.25}$ | 50.75$_{\pm3.72}$ | 52.21$_{\pm1.89}$ | 53.47$_{\pm1.47}$ | 52.9$_{\pm1.36}$ | 53.49$_{\pm1.35}$ | 51.81$_{\pm5.13}$ |
| 4538 | 26.56$_{\pm2.52}$ | 27.69$_{\pm2.5}$ | 26.06$_{\pm0.35}$ | 29.79$_{\pm0.51}$ | 27.04$_{\pm2.62}$ | 27.11$_{\pm2.57}$ | 26.48$_{\pm0.4}$ | 29.26$_{\pm0.65}$ | 23.2$_{\pm1.51}$ | 25.49$_{\pm1.36}$ | 23.92$_{\pm1.27}$ | 22.75$_{\pm4.05}$ |
| 4534 | 54.17$_{\pm3.2}$ | 56.27$_{\pm2.44}$ | 51.86$_{\pm3.32}$ | 52.2$_{\pm2.86}$ | 50.26$_{\pm2.89}$ | 52.55$_{\pm2.77}$ | 49.56$_{\pm3.37}$ | 48.16$_{\pm3.13}$ | 51.8$_{\pm0.71}$ | 51.01$_{\pm1.17}$ | 52.09$_{\pm0.58}$ | 50.2$_{\pm2.97}$ |
| 4134 | 50.8$_{\pm2.06}$ | 51.21$_{\pm2.17}$ | 50.7$_{\pm1.71}$ | 54.54$_{\pm1.8}$ | 50.8$_{\pm3.85}$ | 51.43$_{\pm3.02}$ | 45.84$_{\pm1.99}$ | 51.3$_{\pm1.66}$ | 49.17$_{\pm1.02}$ | 49.44$_{\pm0.99}$ | 48.42$_{\pm0.84}$ | 48.05$_{\pm1.72}$ |
| 40994 | 56.79$_{\pm4.57}$ | 57.63$_{\pm5}$ | 52.25$_{\pm2.2}$ | 53.13$_{\pm3.18}$ | 57.49$_{\pm5.13}$ | 56.15$_{\pm4.32}$ | 50.43$_{\pm1.82}$ | 49.45$_{\pm3.26}$ | 54.2$_{\pm2.14}$ | 54.41$_{\pm2.31}$ | 54.39$_{\pm1.92}$ | 53.75$_{\pm4.27}$ |
| 40984 | 61.87$_{\pm1.98}$ | 63.04$_{\pm2.03}$ | 62.86$_{\pm1.18}$ | 64.57$_{\pm2.04}$ | 62.27$_{\pm1.45}$ | 63.6$_{\pm2.17}$ | 60.59$_{\pm1.24}$ | 62.17$_{\pm1.89}$ | 59.21$_{\pm1.66}$ | 59.61$_{\pm1.54}$ | 58.97$_{\pm1.42}$ | 59.62$_{\pm2.02}$ |
| 40978 | 71.28$_{\pm5.09}$ | 70.7$_{\pm5.1}$ | 72.08$_{\pm5.63}$ | 73.02$_{\pm3.42}$ | 73.44$_{\pm5.65}$ | 71.97$_{\pm5.42}$ | 71.25$_{\pm5.29}$ | 71.94$_{\pm3.58}$ | 68.4$_{\pm2.93}$ | 69.7$_{\pm2.22}$ | 69.59$_{\pm2.81}$ | 67.51$_{\pm5.98}$ |
| 40966 | 32.65$_{\pm2.01}$ | 32.28$_{\pm1.96}$ | 30.71$_{\pm1.98}$ | 33.21$_{\pm2.72}$ | 32.86$_{\pm2.53}$ | 33.24$_{\pm1.47}$ | 31.68$_{\pm1.64}$ | 33.59$_{\pm2.38}$ | 31.34$_{\pm0.84}$ | 31.08$_{\pm0.55}$ | 29.89$_{\pm0.61}$ | 28.28$_{\pm4.88}$ |
| 40701 | 52.19$_{\pm4.35}$ | 53.62$_{\pm4.28}$ | 52.75$_{\pm1.83}$ | 54.76$_{\pm1.26}$ | 52.38$_{\pm3.37}$ | 52.46$_{\pm3.08}$ | 49.87$_{\pm2.06}$ | 55.4$_{\pm1.29}$ | 51.13$_{\pm1.62}$ | 48.49$_{\pm2.33}$ | 48.63$_{\pm1.96}$ | 47.43$_{\pm4.62}$ |
| 3 | 52.49$_{\pm2.69}$ | 53.95$_{\pm2.79}$ | 51.43$_{\pm3.86}$ | 53.83$_{\pm3.17}$ | 54.51$_{\pm2.99}$ | 52.72$_{\pm2.15}$ | 51.13$_{\pm3.64}$ | 52.98$_{\pm3.15}$ | 49.04$_{\pm1.45}$ | 51.16$_{\pm1.34}$ | 50.01$_{\pm1.59}$ | 47.61$_{\pm3.33}$ |
| 38 | 50.46$_{\pm4.48}$ | 55.21$_{\pm4.45}$ | 52.91$_{\pm1.86}$ | 54.77$_{\pm0.64}$ | 54.56$_{\pm4.53}$ | 53.52$_{\pm4.3}$ | 48.76$_{\pm2}$ | 55.44$_{\pm0.23}$ | 48.47$_{\pm2.16}$ | 47.46$_{\pm1.63}$ | 46.84$_{\pm2.05}$ | 48.1$_{\pm5.65}$ |
| 32 | 68.35$_{\pm2.26}$ | 70.15$_{\pm2.33}$ | 67.29$_{\pm1.67}$ | 65.9$_{\pm1.52}$ | 67.95$_{\pm2.63}$ | 70.63$_{\pm2.07}$ | 67.06$_{\pm1.41}$ | 68.17$_{\pm1.23}$ | 66.45$_{\pm1.08}$ | 66.5$_{\pm1.11}$ | 67.28$_{\pm0.98}$ | 64.75$_{\pm4.28}$ |
| 31 | 53.89$_{\pm2.93}$ | 55.24$_{\pm2.88}$ | 55.17$_{\pm1.63}$ | 56.98$_{\pm1.66}$ | 51.92$_{\pm3.78}$ | 53.51$_{\pm2.58}$ | 55.94$_{\pm1.63}$ | 52.07$_{\pm1.51}$ | 50.11$_{\pm1.49}$ | 52.65$_{\pm1.55}$ | 50.06$_{\pm1.79}$ | 51.6$_{\pm4.91}$ |
| 307 | 22.63$_{\pm1.84}$ | 22.06$_{\pm1.74}$ | 23.15$_{\pm1.94}$ | 23.87$_{\pm1.82}$ | 22.4$_{\pm2.92}$ | 21.28$_{\pm2.47}$ | 19.53$_{\pm1.69}$ | 24.77$_{\pm2.31}$ | 19.21$_{\pm1.29}$ | 18.73$_{\pm1.28}$ | 19.26$_{\pm1.43}$ | 18.25$_{\pm2.48}$ |
| 300 | 56.21$_{\pm1.58}$ | 58.91$_{\pm1.67}$ | 58.34$_{\pm1.72}$ | 57.99$_{\pm1.59}$ | 56.08$_{\pm1.41}$ | 57.35$_{\pm1.51}$ | 55.47$_{\pm2.11}$ | 55.66$_{\pm1.31}$ | 54.02$_{\pm1.26}$ | 53.18$_{\pm1.12}$ | 53.11$_{\pm1.27}$ | 51.47$_{\pm2.02}$ |
| 29 | 63.19$_{\pm3.49}$ | 66.09$_{\pm3.58}$ | 63.54$_{\pm3.52}$ | 64.93$_{\pm4.23}$ | 62.23$_{\pm3.54}$ | 66.69$_{\pm3.17}$ | 59.56$_{\pm3.03}$ | 65.23$_{\pm4.31}$ | 60.29$_{\pm1.36}$ | 59.66$_{\pm1.39}$ | 61.7$_{\pm0.75}$ | 59.55$_{\pm5.12}$ |
| 23381 | 50.48$_{\pm2.12}$ | 52.13$_{\pm2.23}$ | 50.38$_{\pm2.27}$ | 51.52$_{\pm1.79}$ | 50.39$_{\pm3.27}$ | 49.8$_{\pm3.14}$ | 49.3$_{\pm2.68}$ | 49.55$_{\pm1.57}$ | 48.19$_{\pm0.63}$ | 49.06$_{\pm0.92}$ | 47.21$_{\pm0.91}$ | 45.57$_{\pm2.02}$ |
| 188 | 28.89$_{\pm2.43}$ | 29.28$_{\pm2.36}$ | 26.32$_{\pm2.42}$ | 28.87$_{\pm3.67}$ | 29.25$_{\pm2.11}$ | 29.84$_{\pm1.23}$ | 27.21$_{\pm2.89}$ | 25.26$_{\pm3.78}$ | 26.93$_{\pm1.13}$ | 27.11$_{\pm1.29}$ | 25.66$_{\pm1.35}$ | 25.41$_{\pm4.45}$ |
| 182 | 64.2$_{\pm3}$ | 67.84$_{\pm3.06}$ | 64.25$_{\pm1.85}$ | 65.16$_{\pm1.13}$ | 65.95$_{\pm2.66}$ | 67.35$_{\pm2.09}$ | 61.29$_{\pm2.22}$ | 65.27$_{\pm1.63}$ | 60.69$_{\pm1.66}$ | 62.06$_{\pm1.85}$ | 61.19$_{\pm1.79}$ | 59.57$_{\pm2.54}$ |
| 1497 | 32.94$_{\pm2.56}$ | 36.55$_{\pm2.47}$ | 36.79$_{\pm1.01}$ | 37.21$_{\pm1.37}$ | 37.91$_{\pm3.09}$ | 36.74$_{\pm2.85}$ | 34.95$_{\pm1.23}$ | 33.59$_{\pm1.12}$ | 30.04$_{\pm1.15}$ | 30.82$_{\pm1.73}$ | 30.06$_{\pm1.08}$ | 30.88$_{\pm2.32}$ |
| 1480 | 51.48$_{\pm2.57}$ | 52.25$_{\pm2.6}$ | 50.17$_{\pm2.83}$ | 50.65$_{\pm1.79}$ | 50.28$_{\pm2.9}$ | 52.61$_{\pm2.83}$ | 49.37$_{\pm2.88}$ | 51.8$_{\pm1.84}$ | 50.25$_{\pm1.14}$ | 47.57$_{\pm1.2}$ | 49.27$_{\pm1.55}$ | 46.48$_{\pm5.46}$ |
| 1478 | 57.33$_{\pm2.56}$ | 57.58$_{\pm2.53}$ | 56.94$_{\pm1.15}$ | 57.33$_{\pm1.6}$ | 58.31$_{\pm2.72}$ | 58.1$_{\pm2.42}$ | 53.42$_{\pm1.01}$ | 57.29$_{\pm1.82}$ | 56.25$_{\pm1.08}$ | 55.66$_{\pm1.14}$ | 56.18$_{\pm1.29}$ | 53.37$_{\pm4.65}$ |
| 1068 | 57.19$_{\pm4.77}$ | 58.81$_{\pm5.01}$ | 56.49$_{\pm3.84}$ | 58.4$_{\pm2.62}$ | 56.96$_{\pm4.73}$ | 59.53$_{\pm4.67}$ | 54.59$_{\pm3.96}$ | 55.89$_{\pm2.37}$ | 54.6$_{\pm2.24}$ | 55.98$_{\pm2.34}$ | 55.91$_{\pm1.83}$ | 54.03$_{\pm5.83}$ |
| 1067 | 61.08$_{\pm4.33}$ | 63.2$_{\pm4.33}$ | 62.51$_{\pm3.34}$ | 63.07$_{\pm3.47}$ | 61.94$_{\pm4.08}$ | 61.02$_{\pm5.13}$ | 60.47$_{\pm3.28}$ | 62.17$_{\pm3.72}$ | 57.82$_{\pm2.26}$ | 58.81$_{\pm2.05}$ | 58.77$_{\pm1.92}$ | 56.3$_{\pm5.34}$ |
| 1063 | 65.85$_{\pm4.32}$ | 67.99$_{\pm4.4}$ | 65.36$_{\pm3.71}$ | 67.41$_{\pm3.96}$ | 65.83$_{\pm4.27}$ | 68.91$_{\pm3.46}$ | 63.98$_{\pm4}$ | 67.06$_{\pm3.66}$ | 63.01$_{\pm1.79}$ | 63.82$_{\pm1.44}$ | 62.45$_{\pm1.58}$ | 63.21$_{\pm4.05}$ |
| 1053 | 52.64$_{\pm4.22}$ | 53.97$_{\pm4.33}$ | 53.98$_{\pm2.8}$ | 55.13$_{\pm2.56}$ | 51.67$_{\pm3.36}$ | 53.83$_{\pm3.05}$ | 53.46$_{\pm2.55}$ | 56.33$_{\pm2.22}$ | 50.28$_{\pm1.84}$ | 50.23$_{\pm1.35}$ | 49.21$_{\pm1.24}$ | 49.55$_{\pm4.37}$ |
| 1050 | 58.28$_{\pm4.5}$ | 61.65$_{\pm4.51}$ | 58.24$_{\pm2.28}$ | 58.94$_{\pm2.2}$ | 58.47$_{\pm3.95}$ | 61.49$_{\pm3.33}$ | 54.16$_{\pm2.16}$ | 58.8$_{\pm2.39}$ | 56.63$_{\pm1.93}$ | 56.49$_{\pm1.98}$ | 56.45$_{\pm2.46}$ | 53.87$_{\pm4.41}$ |
| 1049 | 57.46$_{\pm4.24}$ | 59.85$_{\pm4.5}$ | 59.73$_{\pm2.02}$ | 58.7$_{\pm1.97}$ | 56.27$_{\pm5.05}$ | 58.11$_{\pm4.83}$ | 55.14$_{\pm2.32}$ | 59.15$_{\pm1.54}$ | 54.44$_{\pm1.09}$ | 54.54$_{\pm1.86}$ | 54.47$_{\pm1.88}$ | 55.35$_{\pm5.16}$ |
| 54 | 35.21$_{\pm2.29}$ | 35.59$_{\pm2.38}$ | 35.89$_{\pm2.62}$ | 36.49$_{\pm1.88}$ | 35.7$_{\pm3.11}$ | 36.35$_{\pm3.18}$ | 35.46$_{\pm2.26}$ | 31.94$_{\pm1.48}$ | 33.18$_{\pm1.2}$ | 32.52$_{\pm1.04}$ | 31.9$_{\pm0.82}$ | 31.26$_{\pm4.28}$ |
| 6 | 22.38$_{\pm1.63}$ | 22.2$_{\pm1.55}$ | 24.27$_{\pm1.97}$ | 25.43$_{\pm1.19}$ | 24.28$_{\pm2.53}$ | 26.47$_{\pm1.09}$ | 22.42$_{\pm1.83}$ | 23.05$_{\pm1.01}$ | 20.82$_{\pm1.29}$ | 20.67$_{\pm1.13}$ | 20.15$_{\pm1.18}$ | 17.71$_{\pm2.07}$ |

Table 5: 5-shot test accuracy(%) and standard deviations on 50 datasets from the OpenML-CC18 benchmark, averaged with 100 random seeds.

| | # shot = 5 | | | | | | | | | | | |
|---|---|---|---|---|---|---|---|---|---|---|---|---|
| Id | STUNT | STUNT-P | VIME | VIME-P | SCARF | SCARF-P | SAINT | SAINT-P | LR | KNN | CACTUS | UMTRA |
| 16 | 84.65±1.63 | 86.95±1.71 | 82.63±1.84 | 84.74±1.78 | 81.78±1.78 | 84.65±1.84 | 82.02±2.28 | 83.16±1.3 | 80.75±1.14 | 82.73±1.22 | 80.61±1.07 | 80.6±1.24 |
| 28 | 87.49±1.73 | 91.58±2.21 | 82.4±0.91 | 83.92±0.97 | 82.34±1.07 | 83.9±1.05 | 83.12±0.55 | 83.75±0.57 | 82.19±1.15 | 80.63±1.39 | 81.13±1.26 | 80.73±1.23 |
| 23 | 42.47±3.84 | 42.85±3.26 | 39.26±4.56 | 41.22±3.56 | 39.25±4.2 | 40.52±3.85 | 37.66±4.08 | 38.38±3.49 | 43.58±1.28 | 41.11±1.62 | 40.61±1.24 | 40.85±0.94 |
| 37 | 67.16±6.84 | 66.23±6.52 | 62.43±3.5 | 65.34±2.29 | 64.34±4.76 | 65.7±3.54 | 60.32±3.64 | 64.57±2 | 62.02±4.48 | 64.96±4.41 | 61.29±3.86 | 61.57±4.58 |
| 1501 | 73.48±3.11 | 74.91±3.3 | 64.04±1.62 | 68.13±1.69 | 62.24±1.67 | 69.48±1.69 | 64.39±1.9 | 65.42±1.99 | 63.46±0.95 | 64.57±1.07 | 62.89±1.02 | 63.09±0.84 |
| 40670 | 80.14±2.62 | 83.52±2.36 | 68.5±3.72 | 70.88±3.3 | 69.06±2.9 | 72.68±3.02 | 69.29±4.03 | 68.34±3.48 | 62.85±1.44 | 58.56±1.51 | 60.95±1.24 | 59.97±1.23 |
| 46 | 42.72±5.17 | 46.22±5.41 | 41.73±3.07 | 45.67±2.9 | 44.93±2.68 | 47.25±2.29 | 39.34±3.25 | 46.18±2.66 | 41.18±2.54 | 39.55±2.55 | 40.03±2.71 | 38.27±2.65 |
| 40499 | 73.1±2.19 | 73.79±2.22 | 70.61±1.17 | 74.11±0.94 | 70.27±1.33 | 73.56±1.29 | 68.48±1.54 | 73.15±1.13 | 70.12±0.81 | 69.49±0.92 | 67.57±0.89 | 69.36±1.13 |
| 1475 | 20.05±4.22 | 24.67±3.66 | 19.56±3.97 | 21.52±3.76 | 19.13±4.31 | 22.53±4 | 20.25±3.77 | 19.55±3.82 | 19.77±2.14 | 19.56±1.65 | 19.5±2.17 | 19.45±1.59 |
| 40982 | 45.11±2.81 | 46.02±3.28 | 41.05±2.53 | 42.24±2.55 | 42.51±2.44 | 42.98±2.3 | 40.29±2.74 | 39.74±2.05 | 40.17±1.37 | 41.56±1.58 | 40.76±1.81 | 37.53±1.4 |
| 458 | 95.89±1.23 | 99.61±2.06 | 95.71±1.26 | 98.12±1.11 | 94.13±2.05 | 96.85±1.12 | 96.28±1.02 | 97.1±0.81 | 91.36±0.65 | 92.12±0.74 | 90.08±0.64 | 91.87±0.64 |
| 40979 | 90.46±1.54 | 91.41±1.34 | 83.52±1.07 | 85.34±1.01 | 83.07±1.08 | 86.17±1.34 | 80.64±1.45 | 82.66±0.75 | 88.91±1.04 | 88.71±1.87 | 89.1±1.13 | 89.06±1.14 |
| 14 | 66.34±2.43 | 67.77±3.01 | 63.24±1.92 | 64.21±1.96 | 65.5±1.68 | 66.33±1.69 | 62.38±1.97 | 63.46±2.26 | 65±1.02 | 64.19±1.18 | 64.92±1.42 | 64.06±0.97 |
| 12 | 86.89±1.86 | 88.59±1.66 | 83.12±0.86 | 83.73±0.7 | 83.82±1.16 | 86.25±0.79 | 81.89±0.7 | 83.04±0.48 | 85.27±1.38 | 84.89±1.53 | 85.4±1.43 | 84.69±1.4 |
| 22 | 66.59±2.53 | 68.9±2.21 | 61.43±1.87 | 62.91±1.79 | 60.78±2.07 | 61.14±2.17 | 61.78±2.34 | 63.68±2.05 | 64.74±1.37 | 65.07±1.23 | 64.75±1.24 | 64.61±1.11 |
| 1485 | 53.18±4.72 | 53.27±5.35 | 51.22±2.26 | 52.31±2.32 | 52.27±2.4 | 52.28±2.79 | 49.04±1.9 | 51.92±2.49 | 51.21±1.99 | 51.6±2.19 | 51.29±1.85 | 51.64±2.43 |
| 1487 | 65.81±8.9 | 68.09±8.61 | 68.8±5.63 | 70.34±3.99 | 66.54±3.29 | 69.92±4.41 | 66.47±5.2 | 71.28±3.87 | 63.96±6.18 | 64.02±6.63 | 64.39±6.48 | 64.55±6.76 |
| 44 | 84.26±3.47 | 86.39±3.88 | 82.93±3.15 | 83.13±3.62 | 81.23±2.33 | 83.69±2.38 | 82.64±3.02 | 81.37±3.15 | 82.82±0.94 | 83.17±1.34 | 83.17±0.76 | 81.96±0.85 |
| 1494 | 66.77±6.1 | 69.33±6.18 | 65.43±2.71 | 69.05±3.41 | 66.92±6.22 | 68.76±2.84 | 65.99±2.9 | 69.24±3.35 | 65.58±3.19 | 64.77±3.9 | 64.79±3.62 | 65.3±3.72 |
| 1510 | 91.2±2.49 | 93.14±1.21 | 91.96±0.97 | 92.76±1.21 | 91.13±0.92 | 92.06±1.21 | 91.26±1.21 | 91.72±1.43 | 89.9±1.05 | 89.13±1.26 | 90.1±1.06 | 89.49±1.16 |
| 6332 | 67.65±2.75 | 68.31±2.56 | 68.16±1.31 | 68.03±2.09 | 67.87±1.02 | 67.93±2.12 | 67.67±0.96 | 66.06±1.65 | 66.49±1.25 | 65.97±1.72 | 65.93±1.4 | 65.78±1.32 |
| 4538 | 33.86±2.28 | 36.46±2.31 | 36.91±2.4 | 36.87±2.05 | 37.39±1.75 | 35.81±1.22 | 37.17±2.65 | 35.84±2.4 | 31.93±1.01 | 32.07±1.28 | 31.87±1.23 | 31.99±1.05 |
| 4534 | 61.13±2.4 | 61.15±2.59 | 60.49±1.34 | 60.92±2.22 | 60.59±1.02 | 61.61±2.51 | 57.58±1.05 | 58.46±2.67 | 60.12±1.23 | 58.84±0.95 | 59.35±0.98 | 59.9±1.4 |
| 4134 | 61.18±1.91 | 63.1±1.89 | 62.23±2.57 | 65.98±1.84 | 62.62±0.81 | 63.71±1.18 | 61.81±2.67 | 63.54±1.99 | 59.59±1.37 | 59.12±1.53 | 60.11±1.45 | 58.76±1.54 |
| 40994 | 67.51±2.77 | 73.29±3.25 | 68.25±3.18 | 73.93±2.41 | 73.63±3.39 | 73.18±3.28 | 66.55±2.96 | 72.67±2.66 | 65.92±1.17 | 65.89±1.16 | 65.52±1.65 | 65.86±1.49 |
| 40984 | 70.67±1.77 | 72.83±1.86 | 70.24±0.84 | 73.38±1.23 | 72.6±1.35 | 73.6±0.29 | 70.87±0.55 | 72.24±1.15 | 68.83±1.25 | 68.17±1.4 | 69.02±1.35 | 69.11±1.4 |
| 40978 | 84.34±3.49 | 83.29±3.85 | 82.35±1.94 | 83.91±2.75 | 82.76±4.22 | 82.69±3.3 | 80.54±1.61 | 84.12±2.7 | 82.77±0.55 | 82.48±1.14 | 83.29±0.94 | 82.31±1.35 |
| 40966 | 42±2.17 | 46.17±2.09 | 41.74±1.05 | 46.2±0.87 | 45.8±1.58 | 45.31±3.18 | 41.44±1.49 | 43.54±0.57 | 40.12±0.73 | 40.61±0.85 | 40.61±1.01 | 40.92±0.92 |
| 40701 | 59.61±3.68 | 61.7±3.79 | 61.34±1.25 | 65.12±2.3 | 62.47±0.43 | 61.64±2.8 | 60.86±1.66 | 65.74±2.54 | 58.44±0.71 | 57.51±0.91 | 58.01±0.79 | 57.75±0.91 |
| 3 | 58.46±2.39 | 60.42±2.47 | 60.08±2.83 | 60.88±1.83 | 59.96±3.85 | 60.63±1.68 | 59.25±2.71 | 61.82±1.71 | 56.85±1.16 | 56.89±1.38 | 56.92±1.27 | 56.58±1.13 |
| 38 | 66.66±3.03 | 67.9±3.4 | 68.31±2.91 | 68.99±0.7 | 67.75±3.19 | 66.91±2.6 | 68.12±3.13 | 69.98±1.07 | 64.76±0.92 | 65.63±0.87 | 65.5±0.57 | 65.05±0.99 |
| 32 | 79.13±1.44 | 80.19±1.63 | 79.45±1.65 | 80.47±0.41 | 80.32±2.3 | 81.1±2.96 | 76.6±1.36 | 81.16±0.84 | 77.35±1.06 | 76.86±0.92 | 77.46±0.87 | 78.1±0.93 |
| 31 | 60.75±2.67 | 63.88±2.9 | 64.56±1.42 | 63.06±2.78 | 60.27±1.51 | 64.58±2.39 | 62.99±1.09 | 62.78±2.34 | 59.22±1.33 | 58.87±1.37 | 58.76±1.66 | 59.19±1.24 |
| 307 | 33.9±1.96 | 36.22±1.89 | 36.56±3.39 | 35.5±1.7 | 36.34±1.11 | 36.87±0.85 | 35.78±3.17 | 33.86±1.25 | 32.14±1.44 | 31.58±1.43 | 32.34±1.63 | 32.75±1.49 |
| 300 | 67.7±1.37 | 68.52±1.33 | 68.76±1.2 | 67.61±1.71 | 69.43±2.57 | 68.02±0.72 | 68.08±0.98 | 67.76±1.85 | 66.07±1.03 | 65.67±0.78 | 66.11±0.86 | 65.94±0.84 |
| 29 | 74.46±2.22 | 75.17±2.38 | 74.61±1.91 | 75.52±2.38 | 74.27±2.94 | 75.08±1.77 | 75.45±1.85 | 72.93±2.84 | 72.98±0.75 | 72.67±0.81 | 72.54±0.98 | 73.23±0.76 |
| 23381 | 66.25±2.54 | 68.28±2.08 | 68.09±2.5 | 69.06±3.33 | 68.01±2.43 | 68.18±2.96 | 68.39±2.71 | 69.74±2.88 | 64.81±1.15 | 64±1.2 | 64.26±1.42 | 63.85±1.37 |
| 188 | 38.01±2.24 | 39.18±2.36 | 38.39±2.38 | 41.74±1.04 | 39.69±1.41 | 39.89±2.05 | 39.36±1.96 | 40.28±1.12 | 36.17±1.16 | 36.11±1.02 | 36.65±1.2 | 36.03±1.06 |
| 182 | 72.78±1.57 | 74.29±1.71 | 73.44±2.41 | 73.16±1.03 | 74.45±2.61 | 73.88±3.01 | 70.96±2.26 | 70.31±1.18 | 71.74±1.25 | 70.97±0.98 | 71.31±0.99 | 71.36±1.22 |
| 1497 | 41.71±2.17 | 43.59±2.13 | 43.08±2.01 | 45.2±1.56 | 43.26±1.78 | 42.69±2.7 | 42.07±2.03 | 42.95±1.48 | 39.92±0.69 | 39.45±0.77 | 40.16±1.04 | 39.75±0.97 |
| 1480 | 60.3±1.98 | 63.27±2.24 | 63.99±2.26 | 63.78±2.48 | 64.12±3.04 | 64.11±3.49 | 64.17±2.46 | 61.39±2.21 | 58.6±1.57 | 58.97±1.44 | 58.8±1.46 | 58.69±1.67 |
| 1478 | 68.69±1.78 | 69.85±1.75 | 70.6±2.42 | 70.57±2.93 | 68.89±1.71 | 70.3±0.66 | 69.64±2.07 | 69.39±3.38 | 66.95±1.47 | 67.54±1.33 | 66.8±1.45 | 67.51±1.43 |
| 1068 | 71.55±4.45 | 72.33±4.65 | 72.31±3.55 | 72.29±1.9 | 72.04±1.11 | 71.85±2.19 | 70.17±3.56 | 71.24±1.57 | 70.11±1.55 | 69.73±1.65 | 69.96±1.73 | 69.68±1.75 |
| 1067 | 71.43±3 | 73.2±3.19 | 73.84±1.82 | 73.62±3.37 | 71.08±1.81 | 74.08±1.45 | 73.16±1.67 | 72.06±2.95 | 69.56±1.55 | 70.05±1.88 | 69.77±1.93 | 69.01±1.96 |
| 1063 | 74.31±2.77 | 75.45±2.92 | 76.4±3.07 | 76.36±1.94 | 75.08±3.13 | 75.68±2.01 | 73.85±3.03 | 74.42±1.65 | 73.27±1.56 | 73±1.69 | 72.36±1.38 | 72.15±1.74 |
| 1053 | 59.37±3.51 | 60.5±3.53 | 60.55±1.47 | 60.09±1.5 | 58.4±1.13 | 61.98±1.52 | 60.99±1.73 | 59.28±1.47 | 58.03±0.77 | 57.63±0.92 | 57.53±0.82 | 58.23±1.14 |
| 1050 | 66.32±3.69 | 67.37±3.67 | 66.84±1.64 | 67.04±1.13 | 67.85±0.93 | 67.75±1.28 | 64.13±1.68 | 65.24±1.15 | 64.6±1.23 | 65.21±1.07 | 65.24±1.44 | 65.07±1.62 |
| 1049 | 69.81±2.9 | 69.49±3.09 | 69.38±2.82 | 70.07±2.18 | 69.56±3.3 | 69.68±3.31 | 67.29±3.11 | 70.76±2.42 | 67.88±1.54 | 67.45±1.49 | 68.64±1.4 | 68.17±1.64 |
| 54 | 39.32±2.02 | 41.15±2.08 | 41.13±1.92 | 41.6±1.99 | 40.85±2.25 | 43.98±2.99 | 40.87±2.17 | 41.45±1.87 | 37.44±0.57 | 37.63±0.75 | 38.2±0.53 | 37.6±0.81 |
| 6 | 36.9±1.47 | 40.49±1.41 | 41.09±1.26 | 39.59±0.84 | 40.41±2.76 | 40.94±1.74 | 37.42±1.58 | 38.03±0.82 | 35.34±0.99 | 35.4±1.03 | 35.67±1.08 | 35.24±1.05 |

method. All baseline models, with the exception of the meta-learned model, maintained consistency with 1-shot classification. Among the meta-learning models, we chose to compare with STUNT due to its superior performance over CACTUS and UMTRA (Khodadadeh et al., 2019) in accuracy for both 1-shot and 5-shot classification. As depicted in Table 6 and Table 7, our method demonstrates a substantial improvement even on top-performing models such as SCARF pre-training, VIME pre-training, and STUNT. For instance, using karhunen dataset from OpenML, we observed an increase in accuracy from 86.37% to 89.11%, and a reduction in standard deviation from 2.80 to 1.64. Following the application of our method, all four models exhibited improvements in over 70% of the 50 tested datasets compared to their original counterparts.

Table 6: 10-shot test accuracy(%) and standard deviations on 50 datasets from the OpenML-CC18 benchmark, averaged with 100 random seeds.

| Id | STUNT | STUNT-P | VIME | VIME-P | SCARF | SCARF-P | SAINT | SAINT-P | LR | KNN | CACTUS | UMTRA |
|---|---|---|---|---|---|---|---|---|---|---|---|---|
| | | | | | | # shot = 10 | | | | | | |
| 16 | $86.37_{\pm2.8}$ | $89.11_{\pm1.64}$ | $84.05_{\pm1.45}$ | $86.57_{\pm1.15}$ | $86.32_{\pm1.26}$ | $89.2_{\pm1.4}$ | $83.75_{\pm2.26}$ | $85.8_{\pm0.87}$ | $84.98_{\pm1.46}$ | $84.92_{\pm1.74}$ | $85.14_{\pm1.45}$ | $85.79_{\pm1.68}$ |
| 28 | $88.09_{\pm1.91}$ | $90.53_{\pm1.46}$ | $93.69_{\pm0.55}$ | $94.14_{\pm0.64}$ | $91.11_{\pm0.76}$ | $92.9_{\pm0.73}$ | $93.24_{\pm0.6}$ | $94.64_{\pm0.33}$ | $87.32_{\pm1.39}$ | $87.2_{\pm1.45}$ | $86.75_{\pm1.48}$ | $87.43_{\pm1.54}$ |
| 23 | $41.21_{\pm6.84}$ | $42.44_{\pm4.77}$ | $42.31_{\pm3.74}$ | $42.37_{\pm2.73}$ | $42.7_{\pm3.5}$ | $45.28_{\pm3.58}$ | $42.09_{\pm3.12}$ | $41.77_{\pm2.83}$ | $40.29_{\pm4.75}$ | $40.64_{\pm4.23}$ | $39.8_{\pm3.96}$ | $39.78_{\pm4.26}$ |
| 37 | $68.39_{\pm3.13}$ | $71.82_{\pm2.53}$ | $68.59_{\pm2.34}$ | $72.52_{\pm2}$ | $71.47_{\pm1.02}$ | $74.3_{\pm1.16}$ | $68.55_{\pm2.38}$ | $73.33_{\pm1.83}$ | $67.66_{\pm0.95}$ | $67.84_{\pm0.55}$ | $67.48_{\pm0.44}$ | $67.07_{\pm0.96}$ |
| 1501 | $77.1_{\pm3.16}$ | $80.12_{\pm4.24}$ | $79.69_{\pm1.37}$ | $83.6_{\pm1.15}$ | $78.85_{\pm1.56}$ | $79.69_{\pm1.52}$ | $79.88_{\pm1.99}$ | $83.76_{\pm1.47}$ | $76.2_{\pm0.85}$ | $76_{\pm0.44}$ | $76.08_{\pm0.99}$ | $76.45_{\pm0.41}$ |
| 40670 | $84.72_{\pm3.76}$ | $87.87_{\pm3.58}$ | $79.67_{\pm2.39}$ | $85.72_{\pm2.58}$ | $83.84_{\pm1.57}$ | $87.49_{\pm1.82}$ | $80.24_{\pm3.19}$ | $85.76_{\pm2.39}$ | $83.46_{\pm1.06}$ | $83.95_{\pm1.59}$ | $84_{\pm1.42}$ | $83.45_{\pm1.22}$ |
| 46 | $44.97_{\pm3.26}$ | $50.72_{\pm4.31}$ | $46.37_{\pm2.07}$ | $46.83_{\pm2.34}$ | $43.64_{\pm1.67}$ | $48.18_{\pm1.91}$ | $47.26_{\pm3.4}$ | $46.4_{\pm2.17}$ | $43.78_{\pm0.83}$ | $43.56_{\pm0.49}$ | $44.1_{\pm1.14}$ | $44.19_{\pm1.2}$ |
| 40499 | $98.85_{\pm1.42}$ | $99.3_{\pm1.83}$ | $96.22_{\pm0.74}$ | $97.9_{\pm0.64}$ | $98.4_{\pm1.19}$ | $98.99_{\pm0.9}$ | $99.55_{\pm0.63}$ | $98.23_{\pm0.95}$ | $73.21_{\pm0.99}$ | $73.15_{\pm1.03}$ | $73.12_{\pm1.01}$ | $72.74_{\pm0.98}$ |
| 1475 | $74.15_{\pm5.09}$ | $75.28_{\pm3.31}$ | $76.4_{\pm3.07}$ | $77.58_{\pm3}$ | $75.52_{\pm2.54}$ | $76.39_{\pm3.13}$ | $75.45_{\pm3.46}$ | $77.37_{\pm3.03}$ | $22.21_{\pm2.64}$ | $23_{\pm2.85}$ | $22.36_{\pm2.86}$ | $22.6_{\pm2.39}$ |
| 40982 | $23.59_{\pm2.25}$ | $25.99_{\pm2.7}$ | $22.74_{\pm2.33}$ | $26.64_{\pm2.14}$ | $25.33_{\pm1.88}$ | $25.53_{\pm1.54}$ | $21.78_{\pm2.98}$ | $26.66_{\pm2.39}$ | $47.26_{\pm1.18}$ | $47.77_{\pm0.99}$ | $47.82_{\pm1.18}$ | $47.12_{\pm1.2}$ |
| 458 | $98.85_{\pm1.01}$ | $99.3_{\pm0.97}$ | $96.22_{\pm0.69}$ | $97.9_{\pm0.84}$ | $98.4_{\pm0.84}$ | $98.99_{\pm1.98}$ | $97.16_{\pm0.87}$ | $97.17_{\pm0.9}$ | $97.77_{\pm0.69}$ | $97.56_{\pm0.66}$ | $97.57_{\pm0.42}$ | $98.2_{\pm0.48}$ |
| 40979 | $90.53_{\pm0.98}$ | $93_{\pm0.94}$ | $90.01_{\pm0.76}$ | $91.96_{\pm0.87}$ | $89.71_{\pm0.86}$ | $91.57_{\pm0.88}$ | $90.51_{\pm0.9}$ | $92.15_{\pm1.17}$ | $89.16_{\pm0.97}$ | $89.18_{\pm0.82}$ | $89.6_{\pm0.95}$ | $89.8_{\pm0.86}$ |
| 14 | $68.23_{\pm2.6}$ | $69.91_{\pm1.65}$ | $69.77_{\pm1.4}$ | $70.91_{\pm1.36}$ | $68.3_{\pm1.38}$ | $69.63_{\pm1.49}$ | $70.39_{\pm2.37}$ | $71.75_{\pm1.7}$ | $66.98_{\pm1.55}$ | $67.14_{\pm1.43}$ | $67.62_{\pm1.36}$ | $67.47_{\pm1.35}$ |
| 12 | $86.43_{\pm1.3}$ | $89.03_{\pm1.24}$ | $88.45_{\pm0.61}$ | $90.81_{\pm0.61}$ | $87.12_{\pm0.52}$ | $88.29_{\pm0.77}$ | $89.1_{\pm0.64}$ | $90.47_{\pm0.75}$ | $85.2_{\pm0.98}$ | $85.92_{\pm0.75}$ | $85_{\pm0.83}$ | $85.43_{\pm0.71}$ |
| 22 | $69.01_{\pm1.9}$ | $70.34_{\pm1.72}$ | $68.61_{\pm1.39}$ | $70.98_{\pm1.25}$ | $67.29_{\pm1.46}$ | $70.95_{\pm1.61}$ | $69.59_{\pm1.67}$ | $71.6_{\pm1.38}$ | $68.47_{\pm1.33}$ | $68.35_{\pm1.36}$ | $67.95_{\pm1.43}$ | $67.78_{\pm1.47}$ |
| 1485 | $55.05_{\pm4.69}$ | $55.42_{\pm3.95}$ | $55.89_{\pm2.08}$ | $55.91_{\pm2.36}$ | $54.14_{\pm2.08}$ | $54.21_{\pm2.2}$ | $55.88_{\pm2.32}$ | $56.42_{\pm2.07}$ | $53.88_{\pm1.85}$ | $54.02_{\pm2.14}$ | $54.04_{\pm2.41}$ | $54.13_{\pm1.87}$ |
| 1487 | $66.5_{\pm6.27}$ | $70.2_{\pm4.48}$ | $68.9_{\pm3.53}$ | $69.58_{\pm2.86}$ | $67.81_{\pm3.31}$ | $68.56_{\pm2.62}$ | $69.3_{\pm3.82}$ | $68.83_{\pm2.4}$ | $65.55_{\pm4.04}$ | $65.07_{\pm3.91}$ | $65.63_{\pm3.32}$ | $65.31_{\pm3.59}$ |
| 44 | $86.94_{\pm2.1}$ | $87.23_{\pm1.01}$ | $86.1_{\pm2.77}$ | $88.54_{\pm2.98}$ | $86.73_{\pm1.93}$ | $87.34_{\pm1.53}$ | $85.32_{\pm3.22}$ | $88.76_{\pm3.26}$ | $86.31_{\pm1.01}$ | $85.86_{\pm0.95}$ | $86.03_{\pm0.79}$ | $86.12_{\pm0.75}$ |
| 1494 | $70.98_{\pm3.11}$ | $73.96_{\pm2.18}$ | $72.83_{\pm2.28}$ | $73.83_{\pm2.08}$ | $72.88_{\pm2.23}$ | $73.49_{\pm4.59}$ | $72.94_{\pm3.25}$ | $73.14_{\pm2.36}$ | $69.89_{\pm0.72}$ | $69.96_{\pm0.24}$ | $69.97_{\pm0.89}$ | $70.43_{\pm0.89}$ |
| 1510 | $92.88_{\pm2.33}$ | $94.98_{\pm1.31}$ | $92.47_{\pm0.98}$ | $94.07_{\pm1.28}$ | $91.95_{\pm1.16}$ | $93.09_{\pm0.78}$ | $93.01_{\pm1.13}$ | $94.05_{\pm1.2}$ | $91.9_{\pm1.14}$ | $91.57_{\pm1.12}$ | $91.56_{\pm1.27}$ | $91.7_{\pm1.08}$ |
| 6332 | $75.36_{\pm6.09}$ | $75.61_{\pm5.65}$ | $75.72_{\pm2.79}$ | $75.22_{\pm1.62}$ | $75.69_{\pm2.17}$ | $74.91_{\pm1.18}$ | $76.08_{\pm2.1}$ | $75.48_{\pm1.74}$ | $74.86_{\pm4.04}$ | $74.58_{\pm3.69}$ | $73.97_{\pm3.73}$ | $74.64_{\pm3.98}$ |
| 4538 | $42.52_{\pm4.54}$ | $41.82_{\pm5.02}$ | $42.27_{\pm1.52}$ | $42.08_{\pm1.05}$ | $43.03_{\pm2.71}$ | $42.41_{\pm3.14}$ | $41.35_{\pm1.57}$ | $42.88_{\pm1.14}$ | $41.79_{\pm1.62}$ | $41.13_{\pm1.77}$ | $41.22_{\pm1.96}$ | $41.74_{\pm1.75}$ |
| 4534 | $76.99_{\pm3.22}$ | $76.86_{\pm5.58}$ | $77.73_{\pm0.61}$ | $77.83_{\pm2.45}$ | $77.68_{\pm1.39}$ | $77.35_{\pm2.82}$ | $77.36_{\pm2.44}$ | $78.1_{\pm2.03}$ | $76.39_{\pm0.56}$ | $76.4_{\pm0.89}$ | $76.37_{\pm0.61}$ | $75.97_{\pm0.89}$ |
| 4134 | $66.3_{\pm3.29}$ | $67.22_{\pm3.69}$ | $69.73_{\pm2.77}$ | $69.02_{\pm1.09}$ | $68.08_{\pm2.09}$ | $68.03_{\pm1.18}$ | $70.18_{\pm1.99}$ | $68.27_{\pm1.45}$ | $64.92_{\pm0.97}$ | $64.97_{\pm1.29}$ | $65.6_{\pm0.74}$ | $65.05_{\pm0.29}$ |
| 40994 | $77.08_{\pm6.55}$ | $76.71_{\pm11.63}$ | $76.5_{\pm2.51}$ | $77.18_{\pm2.01}$ | $76.51_{\pm1.68}$ | $77.44_{\pm2.66}$ | $77.32_{\pm2.78}$ | $76.66_{\pm2.23}$ | $76.13_{\pm4.45}$ | $76.56_{\pm3.96}$ | $76.27_{\pm4.25}$ | $76.12_{\pm3.64}$ |
| 40984 | $74.67_{\pm2.83}$ | $75.39_{\pm3.43}$ | $76.36_{\pm0.46}$ | $76.28_{\pm0.94}$ | $76.29_{\pm1.24}$ | $75.9_{\pm1.48}$ | $75.49_{\pm0.81}$ | $76.24_{\pm0.64}$ | $73.95_{\pm1.65}$ | $73.7_{\pm1.79}$ | $73.21_{\pm1.82}$ | $73.18_{\pm1.73}$ |
| 40978 | $90.3_{\pm5.83}$ | $89.64_{\pm10.08}$ | $88.84_{\pm2.08}$ | $88.07_{\pm3.02}$ | $90.32_{\pm2.27}$ | $90.41_{\pm0.64}$ | $87.97_{\pm2.69}$ | $87.8_{\pm3.5}$ | $88.96_{\pm3.77}$ | $88.92_{\pm3.17}$ | $89.5_{\pm3.09}$ | $89.47_{\pm3.31}$ |
| 40966 | $49.83_{\pm5.02}$ | $50.75_{\pm4.19}$ | $50.51_{\pm1.34}$ | $50.71_{\pm0.48}$ | $49.61_{\pm0.19}$ | $50.32_{\pm0.92}$ | $49.94_{\pm1.26}$ | $51.29_{\pm0.62}$ | $48.82_{\pm2.88}$ | $49.12_{\pm2.74}$ | $48.84_{\pm2.66}$ | $48.44_{\pm2.26}$ |
| 40701 | $64.98_{\pm12.93}$ | $65.01_{\pm14.1}$ | $63.83_{\pm2.39}$ | $64.29_{\pm1.88}$ | $63.1_{\pm1.14}$ | $63.26_{\pm0.23}$ | $63.26_{\pm1.94}$ | $65.25_{\pm2.03}$ | $63.88_{\pm10.57}$ | $63.71_{\pm10.85}$ | $64.41_{\pm10.06}$ | $63.82_{\pm10.73}$ |
| 3 | $65.2_{\pm4.87}$ | $64.6_{\pm5.45}$ | $64.13_{\pm1.72}$ | $64.07_{\pm1.26}$ | $64.85_{\pm2.18}$ | $65.37_{\pm2.13}$ | $64.44_{\pm2.14}$ | $63.66_{\pm1.14}$ | $64.56_{\pm2.85}$ | $63.9_{\pm1.87}$ | $64.14_{\pm1.87}$ | $64.1_{\pm2.04}$ |
| 38 | $71.8_{\pm6.55}$ | $71.95_{\pm6.53}$ | $72.3_{\pm2.44}$ | $72.34_{\pm2.32}$ | $72.93_{\pm0.71}$ | $72.47_{\pm0.89}$ | $72.54_{\pm0.29}$ | $72.49_{\pm2.22}$ | $70.89_{\pm4.46}$ | $71.15_{\pm4.24}$ | $71.16_{\pm4.36}$ | $70.87_{\pm4.16}$ |
| 32 | $85.51_{\pm1.36}$ | $85.7_{\pm2.25}$ | $84.29_{\pm2.24}$ | $85.12_{\pm1.99}$ | $84.98_{\pm1.56}$ | $85.01_{\pm1.45}$ | $83.58_{\pm1.08}$ | $84.81_{\pm1.62}$ | $84.87_{\pm0.83}$ | $84.56_{\pm0.97}$ | $84.38_{\pm1.05}$ | $84.47_{\pm1.05}$ |
| 31 | $68.61_{\pm5.76}$ | $67.91_{\pm6.36}$ | $69.19_{\pm0.7}$ | $68.7_{\pm1.14}$ | $66.2_{\pm1.74}$ | $67.09_{\pm1.22}$ | $69.36_{\pm3.17}$ | $68.77_{\pm0.91}$ | $67.93_{\pm3.03}$ | $67.81_{\pm2.89}$ | $67.85_{\pm3.49}$ | $67.66_{\pm3.58}$ |
| 307 | $41.01_{\pm3.17}$ | $40.28_{\pm3.35}$ | $39.33_{\pm2.75}$ | $39.51_{\pm2.52}$ | $40.05_{\pm1.25}$ | $40.21_{\pm0.85}$ | $39.94_{\pm1.8}$ | $38.89_{\pm2.57}$ | $40.02_{\pm0.98}$ | $39.69_{\pm0.92}$ | $40.41_{\pm1.09}$ | $40.28_{\pm0.52}$ |
| 300 | $71.83_{\pm1.53}$ | $71.96_{\pm1.48}$ | $72.42_{\pm0.81}$ | $72.77_{\pm1.04}$ | $72.89_{\pm0.29}$ | $72.92_{\pm0.54}$ | $72.65_{\pm2.02}$ | $72.63_{\pm0.66}$ | $70.83_{\pm1.06}$ | $71.15_{\pm1.04}$ | $71.31_{\pm1.18}$ | |
| 29 | $78.67_{\pm4.08}$ | $78.05_{\pm2.5}$ | $76.27_{\pm1.09}$ | $77.17_{\pm3.36}$ | $78.85_{\pm3.26}$ | $78.88_{\pm2.19}$ | $76.96_{\pm2.78}$ | $77.39_{\pm3.3}$ | $77.38_{\pm1.22}$ | $77.69_{\pm1.42}$ | $77.18_{\pm1.92}$ | $78.05_{\pm1.57}$ |
| 23381 | $70.63_{\pm3.71}$ | $71.6_{\pm3.69}$ | $71.75_{\pm1.53}$ | $72.46_{\pm2.04}$ | $71.97_{\pm1.14}$ | $71.49_{\pm0.68}$ | $72.33_{\pm3.45}$ | $73.1_{\pm2.54}$ | $69.5_{\pm1.24}$ | $69.16_{\pm1.07}$ | $69.36_{\pm1.18}$ | |
| 188 | $44.75_{\pm5.12}$ | $43.99_{\pm4.58}$ | $43.65_{\pm2.44}$ | $43.04_{\pm3.03}$ | $43.79_{\pm0.95}$ | $43.46_{\pm1.55}$ | $43.31_{\pm1.18}$ | $43.83_{\pm3.32}$ | $43.92_{\pm2.72}$ | $43.8_{\pm2.76}$ | $43.96_{\pm2.42}$ | $43.52_{\pm2.39}$ |
| 182 | $79.22_{\pm1.51}$ | $79.76_{\pm1.85}$ | $79.38_{\pm2.95}$ | $79.49_{\pm1.72}$ | $79.46_{\pm1.98}$ | $78.8_{\pm1.22}$ | $79.09_{\pm1.45}$ | $79.92_{\pm1.73}$ | $77.78_{\pm1.08}$ | $78.53_{\pm0.99}$ | $78.46_{\pm1.17}$ | $78.05_{\pm1.06}$ |
| 1497 | $45.73_{\pm4.29}$ | $46.31_{\pm4.52}$ | $46.51_{\pm2.26}$ | $46.56_{\pm0.85}$ | $46.11_{\pm2.57}$ | $46.15_{\pm3.35}$ | $45.56_{\pm1.42}$ | $47.09_{\pm1.28}$ | $44.82_{\pm2.27}$ | $45.2_{\pm1.84}$ | $44.3_{\pm1.59}$ | $44.39_{\pm1.68}$ |
| 1480 | $68.91_{\pm3.09}$ | $68.93_{\pm4.11}$ | $68.45_{\pm1.39}$ | $68.9_{\pm2.46}$ | $69.39_{\pm3.29}$ | $68.79_{\pm1.93}$ | $67.58_{\pm2.55}$ | $68.1_{\pm2.39}$ | $68.11_{\pm0.18}$ | $67.47_{\pm1.05}$ | $67.86_{\pm0.51}$ | $67.96_{\pm0.29}$ |
| 1478 | $75.5_{\pm2.55}$ | $75.95_{\pm2.76}$ | $74.55_{\pm2.47}$ | $75.5_{\pm1.67}$ | $75.85_{\pm2.94}$ | $76.35_{\pm1.02}$ | $74.06_{\pm3.35}$ | $75.56_{\pm1.6}$ | $74.6_{\pm1.15}$ | $74.5_{\pm1.22}$ | $74.53_{\pm1.09}$ | $74.16_{\pm1.44}$ |
| 1068 | $76.87_{\pm13.21}$ | $76.19_{\pm13.74}$ | $77.72_{\pm1.74}$ | $77.84_{\pm1.18}$ | $77.92_{\pm2.71}$ | $78.86_{\pm1.85}$ | $78.08_{\pm1.82}$ | $77.15_{\pm1}$ | $75.91_{\pm10.37}$ | $75.75_{\pm10.42}$ | $75.52_{\pm10.47}$ | $76.15_{\pm10.54}$ |
| 1067 | $85.73_{\pm4.05}$ | $84.97_{\pm5.73}$ | $85.09_{\pm3.14}$ | $85.38_{\pm2.99}$ | $85.21_{\pm0.87}$ | $85.2_{\pm2.75}$ | $85.43_{\pm3.36}$ | $85.77_{\pm2.65}$ | $84.9_{\pm1.06}$ | $84.76_{\pm1.11}$ | $84.73_{\pm1.63}$ | $85.22_{\pm1.32}$ |
| 1063 | $78.86_{\pm5.81}$ | $79.07_{\pm4.83}$ | $77.59_{\pm3.44}$ | $78.44_{\pm3.32}$ | $78.66_{\pm2.43}$ | $79.18_{\pm2.38}$ | $78.26_{\pm1.77}$ | $77.95_{\pm2.83}$ | $77.57_{\pm3.49}$ | $78.29_{\pm2.95}$ | $77.63_{\pm3.03}$ | $77.82_{\pm2.9}$ |
| 1053 | $65.35_{\pm8.74}$ | $66.23_{\pm9}$ | $67.23_{\pm1.85}$ | $67.19_{\pm1.63}$ | $65.28_{\pm2.87}$ | $65.67_{\pm2.81}$ | $67.39_{\pm1.13}$ | $68.06_{\pm1.28}$ | $64.7_{\pm5.8}$ | $64.6_{\pm6.09}$ | $64.49_{\pm6.67}$ | $64.63_{\pm5.89}$ |
| 1050 | $71.87_{\pm8.31}$ | $72.68_{\pm7.28}$ | $73.63_{\pm0.39}$ | $73.64_{\pm1.45}$ | $73.06_{\pm0.58}$ | $72.88_{\pm1.75}$ | $73.88_{\pm1.53}$ | $72.93_{\pm1.36}$ | $70.84_{\pm6.23}$ | $71.07_{\pm5.47}$ | $70.57_{\pm6.09}$ | $70.63_{\pm5.83}$ |
| 1049 | $71.85_{\pm4.36}$ | $71.82_{\pm4.74}$ | $72_{\pm1.27}$ | $71.2_{\pm1.43}$ | $72.02_{\pm1.63}$ | $71.6_{\pm3.19}$ | $71.73_{\pm2.47}$ | $70.68_{\pm1.71}$ | $70.93_{\pm1.38}$ | $70.92_{\pm2.03}$ | $71.1_{\pm1.99}$ | $71.12_{\pm1.92}$ |
| 54 | $45.04_{\pm3.97}$ | $44.9_{\pm3.36}$ | $44.52_{\pm1.72}$ | $45.36_{\pm0.53}$ | $46.32_{\pm0.43}$ | $45.56_{\pm0.76}$ | $45.18_{\pm2.27}$ | $45.56_{\pm0.64}$ | $43.62_{\pm1.43}$ | $43.96_{\pm1.29}$ | $44.21_{\pm1.59}$ | $43.83_{\pm1.93}$ |
| 6 | $47.31_{\pm1.99}$ | $46.71_{\pm1.92}$ | $46.08_{\pm2.17}$ | $45.94_{\pm1.22}$ | $46.55_{\pm2.35}$ | $46.67_{\pm1.03}$ | $43.96_{\pm0.54}$ | $45.22_{\pm1.57}$ | $45.98_{\pm1.53}$ | $45.94_{\pm1.46}$ | $46.76_{\pm1.44}$ | $46.04_{\pm1.69}$ |

## B.3 SIGNIFICANCE TEST

We performed statistical significance tests on results in 50 datasets. Positive is those with averaged results better than the original version. Negative means averaged inferior to the original version. For all versions in 1-shot settings, our versions have P <0.01 in Q2. Even in the third quartile(Q3), P is still largely less than 0.05. Thus, our results have clear advantages over the original versions with statistical significance.

For the few with inferior results, P is normally large. For Q2, all variants have P >0.05. The minimal of VIME-P is P=6.98E-02 >0.05. Our results are similar to the original ones in this limited dataset.

Table 7: significance statistics

| Model | | count | Minimum value | Q1(1/4) | Q2(1/2) | Q3(3/4) | Maximum value |
|---|---|---|---|---|---|---|---|
| | | | # shot = 1 | | | | |
| VIME-P / VIME | Positive | 47 | 3.99E-04 | 1.66E-03 | 5.06E-03 | 3.56E-02 | 6.56E-01 |
| | negative | 3 | 6.98E-02 | 9.73E-02 | 9.73E-02 | 9.73E-02 | 2.38E-01 |
| SCARF-P / SCARF | Positive | 41 | 1.34E-04 | 2.92E-03 | 6.17E-03 | 5.92E-02 | 9.11E-01 |
| | negative | 9 | 3.08E-02 | 7.36E-02 | 7.85E-02 | 9.03E-02 | 8.64E-01 |
| SAINT-P / SAINT | Positive | 43 | 2.03E-05 | 2.32E-02 | 1.41E-01 | 3.51E-01 | 4.74E-01 |
| | negative | 7 | 5.81E-03 | 2.81E-02 | 8.62E-02 | 1.36E-01 | 4.10E-01 |
| STUNT-P / STUNT | Positive | 46 | 1.59E-04 | 7.14E-04 | 5.86E-03 | 3.56E-02 | 6.12E-01 |
| | negative | 4 | 2.11E-02 | 2.61E-02 | 4.25E-02 | 4.25E-02 | 9.97E-02 |
| | | | # shot = 5 | | | | |
| VIME-P / VIME | Positive | 37 | 3.21E-04 | 9.83E-04 | 3.46E-02 | 7.02E-02 | 7.21E-01 |
| | negative | 13 | 2.39E-02 | 6.56E-02 | 9.16E-02 | 8.63E-01 | 9.76E-01 |
| SCARF-P / SCARF | Positive | 38 | 2.06E-04 | 6.57E-04 | 4.05E-03 | 5.75E-02 | 6.11E-01 |
| | negative | 12 | 2.56E-03 | 5.49E-03 | 8.16E-02 | 3.87E-01 | 7.18E-01 |
| SAINT-P / SAINT | Positive | 41 | 4.07E-05 | 1.40E-03 | 5.14E-02 | 8.08E-02 | 1.48E-01 |
| | negative | 9 | 3.40E-02 | 8.20E-02 | 1.54E-01 | 4.51E-01 | 7.81E-01 |
| STUNT-P / STUNT | Positive | 47 | 1.26E-04 | 3.85E-04 | 7.69E-03 | 3.73E-02 | 7.87E-01 |
| | negative | 3 | 5.78E-02 | 1.96E-01 | 1.96E-01 | 1.96E-01 | 3.12E-01 |
| | | | # shot = 10 | | | | |
| VIME-P / VIME | Positive | 39 | 2.39E-04 | 2.82E-03 | 3.82E-02 | 5.82E-02 | 8.88E-02 |
| | negative | 11 | 2.32E-02 | 3.86E-02 | 6.81E-02 | 7.65E-02 | 8.59E-01 |
| SCARF-P / SCARF | Positive | 36 | 5.28E-04 | 5.08E-03 | 4.94E-02 | 8.38E-02 | 4.93E-01 |
| | negative | 14 | 2.23E-02 | 4.75E-02 | 6.04E-02 | 7.86E-02 | 7.92E-01 |
| SAINT-P / SAINT | Positive | 34 | 1.09E-04 | 4.08E-03 | 1.87E-02 | 2.69E-02 | 1.59E-01 |
| | negative | 16 | 2.70E-02 | 3.90E-02 | 4.58E-02 | 4.76E-01 | 5.82E-01 |
| STUNT-P / STUNT | Positive | 35 | 3.41E-04 | 4.33E-03 | 4.41E-02 | 7.89E-02 | 8.20E-01 |
| | negative | 15 | 3.11E-03 | 9.23E-03 | 5.07E-02 | 7.21E-02 | 5.15E-01 |

## C MULTI-TASK LEARNING

In this section, we apply our work to the few-shot multi-task learning task in which multiple learning tasks are solved at the same time.

**Dataset:** For Multi-task learning, we use the emotions dataset from OpenML consisting of a variety of audio data with multiple binary labels and 72 numerical features. In particular, the emotion dataset aims to classify the multiple properties: amazed-surprised, happy-please, relaxing-calm, quiet-still, sad-lonely, or angry-aggressive. Because it is multi-labeled, data can have multiple attributes simultaneously, such as amazed-surprised and relaxing-calm audio.

**Baselines:** We apply our method to VIME pre-training, SCARF pre-training, and STUNT. The pre-training of VIME and SCARF are combined with KNN. The baselines we compared are KNN with supervised models and CACTUS (Hsu et al., 2018) with meta-learning models. Because these models, with only one training process, can be adapted to multiple classification tasks. For example, for the MLP model, in the emotions dataset, six models need to be trained for the classification task.

**Results of multi-task learning:** As shown in Table 8, after applying our method, the accuracy of the model has been significantly improved. In the 1-shot classification problem, STUNT-P demonstrates a significant improvement in accuracy for the amazed-surprised classification task compared to the original STUNT. Specifically, it raises the accuracy from 60.94% to 65.43%, reflecting a noteworthy improvement of nearly 6 percentage points. The standard deviation decreases from 12.42 to 11.54.

Table 8: Few-shot multi-task test accuracy (%) and Standard deviation consists of 6 binary classification tasks with 100 random seeds. We apply our method to VIME, SCARF and STUNT. In the table, 'VIME-KNN' and 'SCARF-KNN' refer to the models obtained by combining the pre-training of VIME, the pre-training of SCARF, and K-Nearest Neighbor (KNN). The models we chosen can be adapted to multiple classification tasks with only one training process. ARI means Average Relative Improvements(%).

| Method / Task | amazed-surprised | happy-please | relaxing-calm | quiet-still | sad-lonely | angry-aggressive |
|---|---|---|---|---|---|---|
| | | | # shot = 1 | | | |
| KNN | 54.78±8.09 | 46.43±7.93 | 55.85± 4.36 | 65.55±6.77 | 57.62±3.46 | 59.69±6.44 |
| CACTUS | 59.32±9.31 | 49.73±8.09 | 56.41±10.37 | 65.38±7.07 | 57.93±12.78 | 60.26±14.31 |
| VIME-KNN | 57.27±4.61 | 47.17±7.70 | 56.52±6.93 | 66.36±5.54 | 60.98±5.47 | 60.96±6.90 |
| SCARF-KNN | 58.43±6.68 | 46.88±6.52 | 58.25±7.28 | 69.69±6.48 | 59.96±6.32 | 61.39±5.63 |
| STUNT | 60.94±12.42 | 51.18±7.16 | 59.58±13.29 | 71.34±9.81 | 58.38±14.72 | 63.83± 11.32 |
| VIME-KNN-P | 62.36±5.36 | 48.50±8.09 | 59.79±8.23 | 68.91±4.41 | 62.99±6.34 | 62.12±6.76 |
| SCARF-KNN-P | 63.67±6.54 | 49.96±6.99 | 62.23±6.90 | 73.03±5.32 | 63.43±6.53 | 64.73±6.52 |
| STUNT-P | 65.43±11.54 | 52.53±7.73 | 63.43±13.18 | 79.61±9.51 | 64.16± 14.31 | 65.43±11.37 |
| ARI(%) | 8.41 | 4.01 | 6.36 | 6.74 | 6.33 | 3.28 |
| | | | # shot = 5 | | | |
| KNN | 71.04±2.22 | 51.09±4.13 | 68.11±3.83 | 80.08±2.39 | 67.69±2.20 | 67.42±3.72 |
| CACTUS | 72.32±5.81 | 54.09±6.07 | 61.04±7.82 | 80.35±8.91 | 66.82±7.38 | 70.95±4.36 |
| VIME-KNN | 69.18±3.84 | 52.59±4.12 | 68.24±3.85 | 83.38±2.63 | 68.14±2.52 | 70.61±3.03 |
| SCARF-KNN | 71.89±3.72 | 53.41±4.34 | 69.54±4.57 | 81.93±3.07 | 70.75±2.86 | 70.43±3.95 |
| STUNT | 70.24±6.84 | 55.75±7.27 | 68.53±7.57 | 82.58±5.30 | 67.92±8.82 | 71.70±5.99 |
| VIME-KNN-P | 74.23±4.18 | 56.15±4.39 | 71.61±4.36 | 84.10±2.74 | 69.02±3.07 | 72.81±3.45 |
| SCARF-KNN-P | 73.86±4.20 | 55.87±4.35 | 72.75±4.33 | 83.04±2.53 | 71.21±3.20 | 73.77±3.36 |
| STUNT-P | 74.02±6.79 | 57.35±6.92 | 70.28±6.32 | 85.67±6.16 | 70.24±8.46 | 73.26±5.02 |
| ARI(%) | 5.14 | 4.75 | 4.04 | 1.99 | 1.79 | 3.34 |
| | | | # shot = 10 | | | |
| KNN | 82.89±1.87 | 64.63±2.62 | 74.01±2.22 | 93.93±1.14 | 74.16±1.07 | 81.46±2.4 |
| CACTUS | 83.08±3.45 | 64.98±3.69 | 71.72±4.07 | 89.32±4.21 | 72.78±5.91 | 80.08±0.57 |
| VIME-KNN | 81.48±2.32 | 63.33±2.85 | 73.59±2.27 | 91.44±1.82 | 72.55±2.13 | 80.34±2.83 |
| SCARF-KNN | 82.37±1.38 | 64.28±2.38 | 74.61±2.75 | 92.95±1.19 | 73.34±1.64 | 81.66±2.9 |
| STUNT | 82.19±3.47 | 63.2±3.79 | 73.54±3.22 | 91.46±2.95 | 72.35±3.74 | 82.77±0.74 |
| VIME-KNN-P | 82.91±2.09 | 64.84±1.93 | 74.72±2.61 | 93.75±1.19 | 74.56±1.34 | 81.34±0.59 |
| SCARF-KNN-P | 83.32±2.47 | 65.59±1.47 | 74.16±2.33 | 92.11±1.99 | 73.19±2.66 | 81.85±0.12 |
| STUNT-P | 84.56±4.15 | 66.4±3.18 | 75.94±3.18 | 93.07±3.87 | 73.12±5.36 | 82.06±0.86 |
| ARI(%) | 1.93 | 3.16 | 1.39 | 1.12 | 1.21 | 0.20 |

In the 5-shot classification problem, it enhances the accuracy from 70.24% to 74.02%, marking an impressive increase of nearly 4 percentage points. In terms of the average relative accuracy improvement across the three models, the most pronounced enhancement is observed in the 1-shot scenario for the amazed-surprised classification task, where the average relative accuracy improvement reaches 8.41%. Our variants also significantly outperform other strong baselines. The results clearly show the importance of selecting the appropriate features for self-supervision.

# D ADDITIONAL ANALYSIS EXPERIMENTS

## D.1 OTHER CORRELATION METRICS

In this section, we endeavor to validate the reliability of Pearson's coefficient as a metric to assess the strength of correlation among data features. To enhance the robustness of our investigation, we integrate STUNT with the Spearman and Kendall coefficients (Pranklin, 1974), respectively. From Figure 6, it is evident that Spearman and Kendall exhibit poorer performance. This can be attributed to Spearman's focus on discrete data, where as Kendall is more suitable for datasets with clear distinctions. In contrast, Pearson's correlation coefficient stands out by demonstrating the highest level of accuracy.

## D.2 DIFFERENT GROUPS

We performed the test for probability range difference as described in section 4.4.1. Our conclusion is that the best range should be [0.2, 0.4] with an average of 0.3. For the impacts of the group, we performed the settings with 2, 5, 10 and 20 groups.

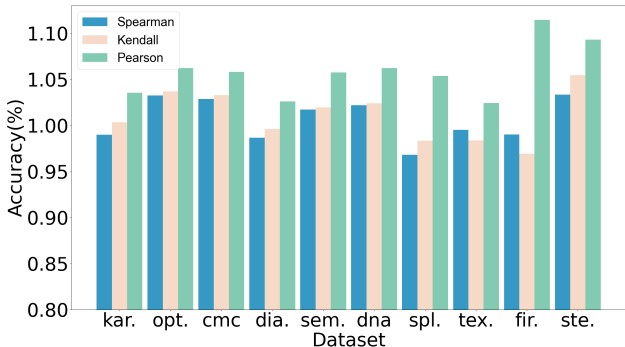

Figure 6: The ratio of the accuracy(revised v.s original) of STUNT with different correlation coefficient metrics on 10 datasets, averaged over 100 random seeds.

Table 9: Dividing into different # groups, 1-shot

| Method / Dataset | karhunen | optdigit | cmc | diabetes | semeion |
|---|---|---|---|---|---|
| | | | # group = 2 | | |
| VIME-P | 63.82±4.42 | 75.84±3.67 | 36.68±5.97 | 56.25±9.84 | 52.63±3.05 |
| | | | # group = 5 | | |
| VIME-P | 64.08±4.01 | 75.47±3.98 | 37.47±6.01 | 57.56±7.37 | 55.2±4.13 |
| | | | # group = 10 | | |
| VIME-P | 66.11±3.65 | 78.07±2.75 | 39.22±4.14 | 58.76±10.19 | 57.39±3.7 |
| | | | # group = 20 | | |
| VIME-P | 67.57±5.26 | 77.05±4.39 | 36.82±4.92 | 56.86±8.29 | 56.99±2.75 |

Table 10: Dividing into different # groups, 5-shot

| Method / Dataset | karhunen | optdigit | cmc | diabetes | semeion |
|---|---|---|---|---|---|
| | | | # group = 2 | | |
| VIME-P | 83.22±1.5 | 82.46±1.27 | 39.58±5.52 | 63.09±4.31 | 64.41±2.29 |
| | | | # group = 5 | | |
| VIME-P | 83.55±1.73 | 83.01±0.51 | 40.92±4.19 | 63.83±4.31 | 65.27±2.45 |
| | | | # group = 10 | | |
| VIME-P | 84.74±1.78 | 83.92±0.97 | 41.22±3.56 | 65.34±2.29 | 68.13±1.69 |
| | | | # group = 20 | | |
| VIME-P | 86.82±1.78 | 82.99±1.91 | 40.59±4.32 | 64.87±4.09 | 67.26±1.94 |

## D.3 PERFORMANCE WITH OTHER STRATEGIES

**Random values for Correlation** We also randomly generate random numbers for each feature as their sum of Pearson coefficients and compare them with the calculated Pearson correlation. In the table below, VIME-P-R, SCARF-P-R and STUNT-P-R denote using random numbers.

It can be observed that, across 10 datasets, their performance is much worse compared to our method. The results are shown in Table 11.

**Other selection strategies:** In our design, we assume that features with higher correlations should be selected more likely for self-supervision. To verify its correctness, we design a new type of variant, VIME-S, STUNT-S and SCARF-S that features are selected in reverse with its correlation order: the higher the correlation, the lower the selection possibility with other settings exactly the same.

Figure 7 shows their performance ratios with respect to the original versions. We can easily see that baselines with reverse correlation selection suffer performance degradation, even worse than the original version, while our design shows significant performance improvements.

Table 11: VIME-P-R, SCARF-P-R, and STUNT-P-R denote replacing Pearson coefficients with random numbers.

| Method / Dataset | karhunen | optdigit | cmc | diabetes | semeion | dna | splice | texture | first | steel |
|---|---|---|---|---|---|---|---|---|---|---|
| | | | | | # shot 1 | | | | | |
| VIME-P-R | 63.74±4.76 | 72.24±2.54 | 37.24±5.37 | 54.53±10.16 | 55.65±3.77 | 53.74±5.61 | 38.17±5.3 | 54.97±2.86 | 15.93±3.69 | 34.36±4.61 |
| SCARF-P-R | 61.4±3.95 | 68.66±4.61 | 35.81±6.09 | 53.78±10.76 | 49.32±4.48 | 49.99±6.82 | 39.77±6.45 | 54.65±4.54 | 16.76±3.15 | 31.95±4.72 |
| STUNT-P-R | 68.93±6.37 | 76.58±5.22 | 36.86±5.76 | 55.76±13.7 | 56.19±7.24 | 68.96±12.08 | 39.66±9.44 | 61.22±5.75 | 17.15±5.82 | 33.53±6.53 |
| VIME-P | 66.16±3.65 | 76.79±2.75 | 39.71±4.14 | 58.9±10.2 | 58.29±3.7 | 57.51±5.81 | 42.31±4.32 | 57.3±2.86 | 18.44±2.57 | 36.91±3.75 |
| SCARF-P | 63.51±3.73 | 71.09±2.94 | 38.49±5.37 | 57.27±8.98 | 54.08±3.9 | 54.44±6.97 | 42.65±4.47 | 59.35±3.91 | 19.03±2.23 | 36.63±4.75 |
| STUNT-P | 73.58±5.82 | 79.83±5.29 | 39.01±5.76 | 60.32±13.34 | 59.66±5.86 | 71.49±11.34 | 43.9±8.08 | 63.89±4.13 | 20.61±4.73 | 37.91±6.14 |
| | | | | | # shot 5 | | | | | |
| VIME-P-R | 82.27±1.9 | 80.92±2.45 | 36.52±4.27 | 60.43±2.49 | 65.99±2.08 | 67.05±4.96 | 42.35±4.22 | 71.15±1.87 | 19.34±3.71 | 39.53±4.39 |
| SCARF-P-R | 82.53±3.28 | 79.97±2.08 | 35.93±3.88 | 62.59±4.68 | 65.99±3.65 | 69.29±3.68 | 42.98±2.54 | 70.6±2.19 | 18.92±4.79 | 40.89±2.28 |
| STUNT-P-R | 82.48±2.84 | 89.31±4.13 | 39.37±3.52 | 63.33±7.3 | 70.89±3.66 | 81.37±3.81 | 43.85±5.8 | 71.12±3.19 | 21.91±4.18 | 42.14±5.17 |
| VIME-P | 84.74±1.78 | 83.92±0.97 | 41.22±3.56 | 65.34±2.29 | 68.13±1.69 | 70.88±3.3 | 45.67±2.9 | 74.11±0.94 | 21.52±3.76 | 42.24±2.55 |
| SCARF-P | 84.65±1.84 | 83.9±1.05 | 40.52±3.85 | 65.7±3.54 | 69.48±1.69 | 72.68±3.02 | 47.25±2.29 | 73.56±1.29 | 22.53±4 | 42.98±2.3 |
| STUNT-P | 86.95±1.71 | 91.58±2.21 | 42.85±3.26 | 66.23±6.52 | 74.91±3.3 | 83.52±2.36 | 46.22±5.41 | 73.79±2.22 | 24.67±3.66 | 46.02±3.28 |

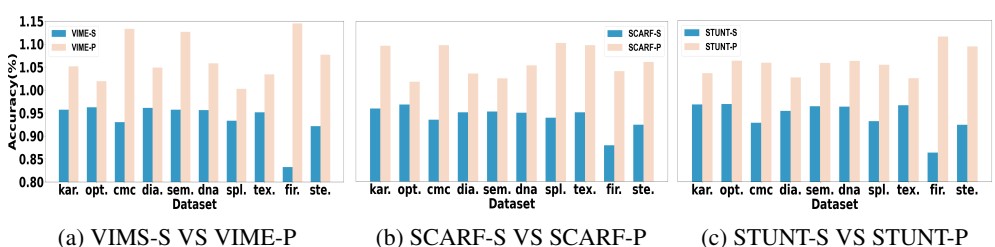

(a) VIMS-S VS VIME-P  (b) SCARF-S VS SCARF-P  (c) STUNT-S VS STUNT-P

Figure 7: Averaged accuracy ratios(revised v.s original) of STUNT with normal/reverse groupings.

## D.4 COMPARISON WITH LLM-BASED TABULAR MODELS

In this section, we compare different LLM-based tabular models as follows:

- P2T: leverages LLMs to extract columns most relevant to the label from the source dataset (e.g., unlabeled data). It then constructs pseudo-examples by treating these columns as prediction targets and using the remaining columns as inputs, generating samples highly aligned with the target task.

- LIFT-ICL: converts data into natural language sentences and forms prompts by concatenating several labeled examples with the test sample. These prompts are then fed into a pretrained language model for direct prediction. k-shot means it contains k labeled examples + 1 test example in the prompt.

- STUNT-LLM: uses LLM(gpt-4.1) to select important features, and then performs STUNT self-supervised learning on these features. STUNT-LLM-70/STUNT-LLM-50 means select 70/50 percent of the most important features sorted by LLM for model training.

Table 12 presents accuracy comparisons of various LLM-based tabular models across 10 datasets under both 1-shot and 5-shot settings. Unlike P2T and LIFT-ICL, which rely on meta-data and label information for tuning, STUNT is fundamentally self-supervised and task-agnostic. STUNT-LLM sometimes struggles, likely because it mixes multiple criteria like correlation and mutual information for selection, making it more sensitive to noise. There are also concerns about potential information leakage in LLM-based methods. For example, P2T requires partial access to the training set during pretraining, which might raise questions about the fairness of the evaluation. That said, since LLMs possess strong knowledge extraction capabilities, an interesting yet important question is how to better leverage these strengths to further enhance methods like ours.

## D.5 ITERATIONS

We adhere to STUNT for evaluation. It is hard to learn the model in the few-shot scenario, especially for the Catboost. CatBoost has a more complex structure than KNN and LR. As shown in Table 1, we perform 100 and 1000 iterations for CatBoost in 1-shot. Catboost achieves the best results with 10K iterations.

Table 12: Accuracy(%) Comparison with different LLM-based tabular models on 10 datasets.

| Method / Dataset | karhunen | optdigit | cmc | diabetes | semeion | dna | splice | texture | first | steel |
|---|---|---|---|---|---|---|---|---|---|---|
| | | | | | # shot 1 | | | | | |
| P2T | 64.75 | 75.63 | 38.06 | 57.08 | 56.59 | 55.69 | 40.69 | 56.25 | 17.33 | 35.71 |
| LIFT-ICL | 63.98 | 73.63 | 37.31 | 54.49 | 54.15 | 53.86 | 40.31 | 53.05 | 15.18 | 33.15 |
| STUNT | 71.06 | 75.15 | 36.87 | 58.79 | 56.41 | 67.31 | 41.67 | 62.38 | 18.49 | 34.68 |
| STUNT-LLM | 68.23 | 71.02 | 34.89 | 56.21 | 55.92 | 64.18 | 40.37 | 60.63 | 17.21 | 32.64 |
| STUNT-LLM-70 | 54.37 | 60.12 | 28.05 | 41.95 | 44.57 | 49.95 | 30.91 | 52.9 | 14.17 | 26.24 |
| STUNT-LLM-50 | 31.34 | 38.59 | 14.67 | 28.94 | 26.82 | 31.65 | 12.67 | 22.95 | 7.6 | 14.88 |
| STUNT-P | 73.58 | 79.83 | 39.01 | 60.32 | 59.66 | 71.49 | 43.90 | 63.89 | 20.61 | 37.91 |
| | | | | | # shot 5 | | | | | |
| P2T | 83.95 | 82.96 | 40.17 | 64.23 | 67.44 | 69.03 | 45.13 | 72.31 | 20.16 | 41.44 |
| LIFT-ICL | 80.76 | 81.82 | 38.87 | 61.74 | 64.56 | 68.36 | 42.52 | 71.53 | 19.34 | 39.17 |
| STUNT | 84.65 | 87.49 | 42.47 | 67.16 | 73.48 | 80.14 | 42.72 | 73.10 | 20.05 | 45.11 |
| STUNT-LLM | 83.41 | 85.16 | 40.73 | 64.15 | 70.38 | 78.39 | 40.61 | 72.52 | 19.47 | 43.38 |
| STUNT-LLM-70 | 63.65 | 70.55 | 34.86 | 46.27 | 51.05 | 61.39 | 32.49 | 59.52 | 14.79 | 34.39 |
| STUNT-LLM-50 | 46.29 | 35.22 | 17.25 | 40.31 | 40.69 | 54.24 | 12.63 | 46.71 | 9.87 | 26.2 |
| STUNT-P | 86.95 | 91.58 | 42.85 | 66.23 | 74.91 | 83.52 | 46.22 | 73.79 | 24.67 | 46.02 |

Table 13: different iteration

| Dataset | karhunen | optdigit | cmc | diabetes | semeion | dna | splice | texture | first | steel |
|---|---|---|---|---|---|---|---|---|---|---|
| 100 iter | 19.76±7.88 | 18.64±5.3 | 13.56±4.78 | 18.17±6.57 | 16.33±10.33 | 10.77±4.39 | 12.86±6.89 | 14.8±4.64 | 5.86±5.74 | 11.32±9.16 |
| 1k iter | 37.36±6.39 | 42.14±5.86 | 26.92±5.96 | 38.83±17.02 | 30.72±7.04 | 31.15±11.16 | 27.88±7.84 | 40.06±5.82 | 11.29±6.69 | 24.44±8.91 |
| 10k iter | 55.15±3.45 | 61.56±3.03 | 38.75±3.66 | 55.28±11.16 | 41.98±4.94 | 43.6±8.57 | 40.46±5.6 | 56.45±2.39 | 17.33±1.84 | 34.11±3.94 |

## D.6 T-SNE Distributions

We visualize embeddings for the test set of the *semeion* dataset with t-SNE (Laurens & Hinton, 2008) and calculate the clustering accuracy (ACC), normalized Maximal Information Coefficient (NMI) with Figure 8. This Figure shows that adopting our strategy can significantly improve the quality of embeddings with much higher results in ACC and NMI. Thus, it can partially explain why our proposed strategy can increase classification performance.

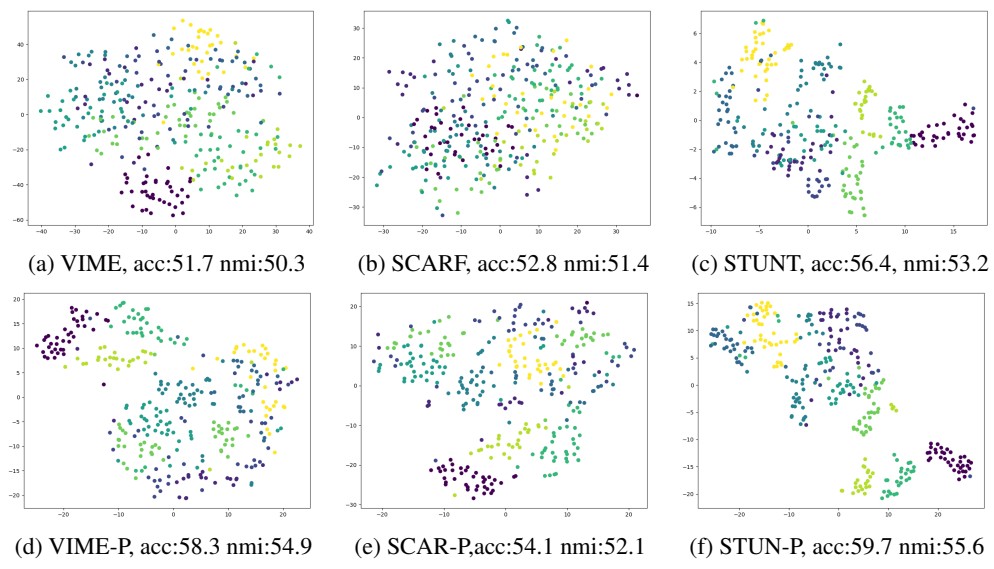

(a) VIME, acc:51.7 nmi:50.3    (b) SCARF, acc:52.8 nmi:51.4    (c) STUNT, acc:56.4, nmi:53.2

(d) VIME-P, acc:58.3 nmi:54.9    (e) SCAR-P,acc:54.1 nmi:52.1    (f) STUN-P, acc:59.7 nmi:55.6

Figure 8: t-SNE Plot of semeion with different models.

Notably, Figure 8 models demonstrate improved performance in classifying the *semeion* dataset under the conditions defined by the 1-shot classification scenario.

## D.7 CATBOOST DEPTH ANALYSIS EXPERIMENT

Regarding the hyperparameter settings of supervised strong baseline, we actually follow the exact same settings as STUNT. Those settings are already optimized for few-shot learning rather than normal learning. To further validate this claim, we have conducted tests for the performance of Catboost with tree depth at 3, 5, and 10. The Table 14 shows the performance of CatBoost is the best, too shallow(e.g. 3,5) or too deep(10), the performance deteriorates. Regardless of the setup, our enhanced model, e.g. STUNT-P has significant performance edge over the Catboost in the few-shot learning on all tested 10 datasets.

Table 14: Campare with Catboost with different depth on 10 datasets

| Method / Dataset | karhunen | optdigit | cmc | diabetes | semeion | dna | splice | texture | first | steel |
|---|---|---|---|---|---|---|---|---|---|---|
| | | | | | # shot 1 | | | | | |
| CatBoost(3) | 52.31±2.89 | 58.62±5.23 | 36.29±5.25 | 52.72±12.91 | 39.1±6.82 | 40.64±9.83 | 37.62±4.25 | 53.64±3.41 | 15.09±1.33 | 31.53±2.59 |
| CatBoost(5) | 53.65±3.96 | 59.63±3.05 | 37.45±4.47 | 53.28±9.93 | 40.35±6.41 | 41.86±9.26 | 38.91±5.18 | 54.73±1.38 | 16.29±2.67 | 32.28±3.72 |
| CatBoost(8) | 55.15±3.64 | 61.56±2.68 | 38.75±2.8 | 55.28±10.99 | 41.98±6.44 | 43.6±8.03 | 40.46±6.76 | 56.45±1.44 | 17.33±0.94 | 34.11±6.17 |
| CatBoost(10) | 53.22±3.45 | 59.44±3.03 | 36.32±3.66 | 53.37±11.16 | 39.98±4.94 | 41.26±8.57 | 38.08±5.6 | 54.74±2.39 | 16.23±1.84 | 32.13±3.94 |
| STUNT-P | 73.58±4.28 | 79.83±1.66 | 39.01±3.79 | 60.32±10.32 | 59.66±5.81 | 71.49±8.16 | 43.9±6.32 | 63.89±2.47 | 20.61±0.99 | 37.91±3.82 |
| | | | | | # shot 5 | | | | | |
| CatBoost(3) | 78.2±1.29 | 79.23±1.38 | 41.11±1.58 | 59.5±4.27 | 61.09±1.34 | 60.81±1.66 | 38.86±2.19 | 67.23±0.85 | 17.47±2.32 | 37.53±1.58 |
| CatBoost(5) | 79.14±1.08 | 80.87±0.77 | 41.59±0.98 | 60.8±4.66 | 61.68±0.96 | 61.07±1.45 | 39.5±2.42 | 68.82±1.2 | 17.96±2.63 | 38.4±1.31 |
| CatBoost(8) | 80.75±1.14 | 82.19±1.15 | 43.58±1.28 | 62.02±4.48 | 63.46±0.95 | 62.85±1.44 | 41.18±2.54 | 70.12±0.81 | 19.77±2.14 | 40.17±1.37 |
| CatBoost(10) | 78.98±1.59 | 80.5±1.46 | 42.38±1.15 | 60.93±4.29 | 61.2±1.32 | 61.31±1.32 | 39.62±2.77 | 68.34±0.65 | 18.35±1.86 | 39.4±1.62 |
| STUNT-P | 86.95±1.71 | 91.58±2.21 | 42.85±3.26 | 66.23±6.52 | 74.91±3.3 | 83.52±2.36 | 46.22±5.41 | 73.79±2.22 | 24.67±3.66 | 46.02±3.28 |

# E   PERFORMANCE IN DATASET WITH CERTAIN CHARATERISTICS

In this section, we conducted various experiments on datasets with various extreme features to verify the robustness of our proposed method.

**Datasets with weakly correlated features:** In this section, we want to explore the feasibility of our method in datasets with weakly correlated features, often found in highly heterogeneous datasets. Three datasets are selected in OpenML with the id. 40499, 407801 and 23381. As shown in Table 15, the correlation of all features is around 0.1.

Table 15 shows the their Pearson correlation distribution and performance comparison with / without our correlation-based enhancements. The results clearly show that our method can still effectively improve existing SSL methods on these datasets. The reason for this improvement is that those features with relatively higher correlations, albeit comparablely low, can still provide more information during self-supervisions than those features with very small correlations. These results clearly show the robustness of our proposed results.

Table 15: Accuracy(%) on datasets with weakly correlated features. The second line means the minimum and the maximum Pearson coefficient for each feature in the dataset. ARI(%) means Average Relative Improvements(%).

| Dataset id | 40499 | 40701 | 23381 |
|---|---|---|---|
| Coefficient | (0.061 0.067) | (0.056 0.111) | (0.123 0.147) |
| STUNT | 62.38 | 52.19 | 50.48 |
| VIME | 55.43 | 52.75 | 50.38 |
| SCARF | 54.06 | 52.38 | 50.39 |
| STUNT-P | 62.89 | 53.62 | 52.13 |
| VIME-P | 57.30 | 54.76 | 51.52 |
| SCARF-P | 59.35 | 52.46 | 49.8 |
| ARI(%) | 2.80 | 3.15 | 2.92 |

**Synthetic dataset with nonlinear and non-monotonic relations:** As Pearson correlation criteria is often seen to measure the strengh of linear relations, here we want to check whether it can be applicable in datasets with non-linear and non-monotonic relations. We draw $x$ from a normal distribution with the following relationships: $x_1 = \sin x, x_2 = -x^2, x_3 = e^x, x_4 = x^3$. Here, we constructed a relatively simple dataset defined as $y = x_1 + x_2 + x_3 + x_4$, where we assign a class label of 1 if $y > 0$, otherwise 0. Here, 1000 samples are generated. In this dataset, $x_1$ and $x_2$ have non-linear and non-monotonic relations, $x_1$ has non-linear relations with $x_3$ and $x_4$.

Table 16 shows the comparison with pure-randomness and correlation-based randomness. As can be seen, even in datasets with non-linear, our method still shows improvements compared to the random method, as we do not need to accurately evaluate the strength of correlations. Instead, our group-based methods enable us to have a rough estimate of overall strengths. Furthermore, the Pearson correlation metric also focuses on the simple linear relations which is crucial for late self-supervised model learning. We guess that is why the Pearson-based evaluation can also effectively work in datasets with nonlinear and non-monotonic relations.

Table 16: Accuracy(%) on the synthetic dataset with nonlinear and non-monotonic relations. The second line denotes Average Relative Improvement(%).

| STUNT | STUNT-P | VIME | VIME-P | SCARF | SCARF-P |
|---|---|---|---|---|---|
| 68.31 | 70.59 | 68.41 | 69.27 | 68.29 | 70.13 |
| - | 3.33 | - | 1.25 | - | 2.69 |

**Noisy data and unbalanced datasets:** In this section, we want to check our methods on datasets with noisy data. Those datasets are inserted with three different types of noise: the *zero-masking* noise: put 0 on the selected value, the *Gaussian* noise: put Gaussian value to the selected value, and the *marginal distribution* noise by replacing the value with a column-wise randomly selected value. We conducted tests under 1-shot and 5-shot settings on the dataset obfuscated with noise. The results with original STUNT and *-P are put before and after /.

We can easily see from Table 17, our method can still enhance the performance of STUNT on different datasets. Masking with the marginal distribution achieved the most significant gain with our proposed method. It is interesting to see that some performance gains are achieved with 30% noise, especially for the marginal distribution. One of our working direction is to analyze the impacts from the types and strength of different noise.

Table 17: Accuracy(%) with original / ours model on datasets with obfuscated with different noise in STUNT. Noise is introduced to 30% values randomly selected from all samples.

| Dataset | No-masking | Zero-masking | Gaussian | Marginal dis. |
|---|---|---|---|---|
| | | # 1 shot | | |
| dibetes | 56.43/58.07 | 56.92/58.79 | 57.57/59.31 | 58.79/60.32 |
| cmc | 35.17/38.24 | 35.83/38.64 | 34.92/37.86 | 36.87/39.01 |
| | | # 5 shot | | |
| dibetes | 65.28/65.74 | 66.76/67.83 | 65.94/66.03 | 67.16/67.42 |
| cmc | 41.79/42.18 | 40.62/41.39 | 41.29/42.36 | 42.47/42.85 |

We also check the performance with our solutions on unbalanced data with seven datasets from the OpenML dataset. STUNT-P and VIME-P consistently maintain performance edges compared to the original version. It shows our solution can be effective in datasets with different distributions, which can be partially reflected in the imbalanced number of samples of different label types.

Table 18: Accuracy(%) on unbalanced datasets. Ratio for the number of samples for each class.

| Dataset id | Ratio | STUNT | STUNT-P | VIME | VIME-P |
|---|---|---|---|---|---|
| 40701 | 6:1 | 52.19 | 53.62 | 52.75 | **54.76** |
| 1487 | 14:1 | 56.72 | **57.73** | 51.33 | 54.86 |
| 46 | 2:1:1 | 41.67 | **43.9** | 42.21 | 42.31 |
| 23 | 2:2:1 | 36.87 | 39.01 | 35.04 | **39.71** |
| 1475 | 5:2:1:1:1:1 | 18.49 | **20.61** | 16.11 | 18.44 |
| 37 | 2:1 | 58.79 | **60.32** | 56.15 | 58.9 |
| 40670 | 2:1:1 | 67.31 | **71.49** | 54.37 | 57.51 |