# OpenReview forum: "Correlation-based Self-Supervision for Few-shot Tabular Learning"
_ICLR.cc/2026/Conference — Submitted to ICLR 2026_

### Official Review · Reviewer_uhuU · 2025-10-23

**Soundness:** 3
**Presentation:** 3
**Contribution:** 2
**Rating:** 4
**Confidence:** 4

**Summary:**

The paper proposes a correlation-guided feature selection strategy to replace uniform random sampling in self-supervised tabular learning.

**Strengths:**

- The paper is clearly written and demonstrates comprehensive and reliable experimental results across diverse baselines and datasets.
- The paper is well-motivated, illustrated in Figure 1, where the degradation from unrelated features highlights the weakness of uniform feature selection in self-supervised tabular learning.

**Weaknesses:**

1. Lack of analysis on why correlation-based selection works

While the proposed correlation-guided feature selection consistently improves performance across SSL baselines, the paper lacks a clear theoretical or mechanistic explanation of why it works.
The improvement is shown empirically (Sections 4.3–4.5), but there is no analysis linking correlation strength to pseudo-label informativeness, representation consistency, or optimization stability.
It remains unclear whether the gains stem from mutual-information preservation, noise filtering, or redundancy reduction.
Without such analysis, the method remains an effective heuristic rather than a principled approach.

2. Incomplete handling of categorical features

The proposed framework processes categorical features using simple encodings (e.g., one-hot or integer), treating all resulting columns equally when computing correlations. However, one-hot encoded columns are linearly dependent and often exhibit artificial negative correlations that do not capture true semantic relations among categories. Consequently, the correlation-based weighting may misrepresent the contribution of categorical features, particularly in datasets with many discrete attributes. The paper acknowledges this simplification but does not analyze its impact or provide strategies to mitigate the resulting bias.

**Questions:**

W1, W2

---

> ### Author Response · Authors · 2025-12-03
>
> A4.1: If it is redundant information, from the perspective of information theory, this information is actually noise, which will have an adverse impact on the performance of the model. Our method can capture this point. In Appendix E, we have provided relevant data. As the noise increases, the performance of the model will decline. However, compared with the original model, our method still has an improvement.
>
> We think this solution is novel albeit its simplicity.  We actually raise a simple but fundamental question on how to perform self-supervision and when self-supervision is not effective. To the best of our knowledge, all tabular supervision solutions adopt randomness as the general guidelines of pseudo-label generation and largely ignore the complex relations among features. They generally assume that features are of the same importance. This paper changes this situation. Although this discussion is limited to tabular data, we are working to apply it to other data types with encouraging results. Thus, we think it is novel and important.
>
> From theory perspectives, we point out that if features are un-related. The self-supervision is harmful which have not been pointed out before although we only use basic information theory. Of course, due to lack of labels and the diversity of different datasets, we do not have any metric to evaluate the quality of  pseudo labels. Lack of those preliminary preparation make it hard to build those links. To the best of our knowledge, there no such metrics exists in the whole self-supervision domain.
>
>
>
> A4.2: The question you raised is very good, but it is not what we are discussing in this article. What we focus on is guiding the generation of self-supervised models based on the correlations among features to enhance the performance of the models. The models involved in our article include VIME and STUNT, both of which have been tried with different encodings. Of course, we also tried Tab-Rep[1]. As shown in the table below, after using One-hot and Tab-Rep respectively in the STUNT model and applying our method, the comparison of model performance shows that our method can also bring performance improvement on Tab-Rep.
>
>
> | |STUNT|Pearson|
> |:-|:-:|:-:|
> |One-hot| 58.79 / 60.32  | 67.31 / 71.49 |
> |Tab-Rep|   57.24 / 59.47 	|  67.12 / 70.34  |
>
>
> [1]Si J, Ou Z, Qu M, et al. TabRep: Training Tabular Diffusion Models with a Simple and Effective Continuous Representation[J]. arXiv preprint arXiv:2504.04798, 2025.

---

### Official Review · Reviewer_HWDA · 2025-10-30

**Soundness:** 2
**Presentation:** 2
**Contribution:** 2
**Rating:** 4
**Confidence:** 3

**Summary:**

The paper presents a novel approach to enhance self-supervised learning for few-shot tabular data.

The research addresses the significant difficulty in applying self-supervised learning to tabular data especially within a few-shot learning context. The fundamental challenge arises from the intrinsically heterogeneous nature of tabular data, which severely hampers the generation of high-quality pseudo-labels. The author's purpose is to propose a simple yet effective solution: to utilize the correlation among original features to qualitatively evaluate the possible quality of pseudo-labels, moving beyond the current homogeneous assumption in tabular learning. The goal is to develop a correlation-based randomness approach to select the most informative features for pseudo-label generation.

Beforehand existing self-supervision approaches for tabular data rely on a random-based selection methodology for generating pseudo-labels or augmented views. This common design suffers from treating different features equally, effectively ignoring the complex and unequal relationships among features. This uniform random selection might induce substantial noise into the self-supervision process, particularly when the selected features have little relation to others. The paper experimentally shows that the efficacy of self-supervision rapidly deteriorates with the introduction of certain unrelated features. For example, introducing just two unrelated features can result in a twenty percent accuracy loss in both the diabetes and cmc datasets.

The core idea is to introduce Correlation-based Randomness to guide the feature selection for pseudo-label generation. It first calculates the averaged absolute values of the correlation coefficients between a feature and all other features to determine its overall association strength. Features with greater overall correlation are assigned a higher probability of being selected. This is logically supported by the notion that features highly correlated with others contain more important information for self-supervision. To mitigate the influence of out-of-distribution or erroneous coefficients, the method employs a group-based selection strategy. Features are first sorted by their average correlation and then equally divided into groups, with the selection probability increasing linearly across the groups. The author argues that this approach works well because it prioritizes the most informative features, which aligns with information theory principles that suggest pseudo-labels independent of inputs constitute noise and result in low-quality models.

The author verifies the method's performance by applying the proposed strategy to four representative self-supervision baselines for tabular data: VIME, SCARF, STUNT, and SAINT. The enhanced variants, denoted with a -P suffix, are evaluated in one-shot, few-shot, and multitask few-shot tasks across more than fifty OpenML datasets. The experimental results reveal significant performance improvements. Specifically, the variants achieve a performance gain on over ninety percent of the evaluated datasets in the one-shot learning scenario, and approximately eighty percent in the five-shot learning contexts. These enhancements are further substantiated by a statistical significance test demonstrating a clear advantage over the original versions. For example, in one-shot learning, VIME-P achieved an average improvement of three point forty-four percent.

The conclusion is that utilizing feature correlation provides an effective mechanism to generate higher quality pseudo-labels, leading to substantial and statistically significant performance gains for few-shot tabular learning models. The main contribution of the paper is threefold: first, it rigorously reveals the limitations of the prevailing uniform random feature selection method in self-supervision for tabular data, showing it introduces significant noise. Second, it proposes a Correlation-based Randomness design, which is a simple and generalizable technique to integrate feature relations into the pseudo-label generation process. Third, it provides outstanding experimental results demonstrating the effectiveness and robustness of the proposed strategy across diverse baselines and datasets.

**Strengths:**

The method directly attacks the core logical weakness of current self-supervised tabular methods, which is the assumption that all features contribute equally to the self-supervision task. By introducing Correlation-based Randomness, it ensures that the feature selection process is no longer noise-prone and instead prioritizes features with a high dependency on others, which is critical for generating informative pseudo-labels.

The key contribution is providing a simple yet universally applicable design that can be easily integrated into diverse existing self-supervision frameworks, including generative, contrastive, and meta-learning models. This demonstrated generalizability is a major strength.

**Weaknesses:**

The logical explanation is not entirely sufficient regarding the choice of the best metric. While eight correlation metrics are tested, and Pearson correlation is shown to be superior, the justification is largely empirical and post-hoc, attributing its success to its ability to capture linear and non-linear monotonic associations. A deeper theoretical argument is needed to explain why these specific types of associations are fundamentally more critical for few-shot pseudo-label generation than the more general association measures offered by other tested metrics like Maximal Information Coefficient or Distance Correlation.

The paper provides robust evidence by testing the proposed strategy on four diverse baselines across over fifty datasets, with statistical confidence reported for one-shot and five-shot scenarios. However, the scope is primarily few-shot classification. Validation across other few-shot tabular learning tasks, such as regression or time-series forecasting, is not provided and would strengthen the claim of universal applicability.

A concern point is the method's raw reliance on standard correlation metrics. The paper notes that for categorical features, no specific embedding operation is applied. Standard correlation coefficients, such as Pearson, are fundamentally designed for continuous variables. Using them directly on categorical or mixed-type features without a dedicated, domain-aware embedding or a mixed-data specific correlation metric might compromise the logical foundation for certain datasets and limit the optimality of the feature selection. Another concern is the unavoidable pre-computation overhead for the correlation matrix, which, despite being argued as negligible because it is done offline, still introduces an extra computational step that may be non-trivial for very large or rapidly changing datasets.

**Questions:**

Given the demonstrated empirical success of Pearson correlation, what is the precise theoretical mechanism that makes its measure of non-linear monotonic associations more impactful for feature informativeness in the few-shot self-supervision context compared to the general association measures offered by non-parametric metrics?

How is the Pearson correlation coefficient robustly and optimally calculated in practice for datasets with a significant number of categorical features, considering the paper stated no specific embedding operation was applied to them? Would adopting a more suitable mixed-data correlation metric not provide a more logically sound and powerful basis for feature selection across all tabular data types?

Since the feature selection probability relies on the absolute value of correlation, what is the impact of explicitly ignoring the direction of correlation positive or negative on the self-supervised task, particularly for models like VIME that use reconstruction or SCARF that rely on contrastive learning?

For scenarios involving massive or streaming tabular datasets where continuous updates or huge pre-computation is a barrier, how can the relatively efficient but still significant O of m squared n complexity of the Pearson correlation calculation be adapted for efficient online or incremental updating of the feature selection probabilities?

---

> ### Author Response · Authors · 2025-12-03
>
> A3.1: We think this solution is novel albeit its simplicity.  We actually raise a simple but fundamental question on how to perform self-supervision and when self-supervision is not effective. To the best of our knowledge, all tabular supervision solutions adopt randomness as the general guidelines of pseudo-label generation and largely ignore the complex relations among features. They generally assume that features are of the same importance. This paper changes this situation. Although this discussion is limited to tabular data, we are working to apply it to other data types with encouraging results. Thus, we think it is novel and important.
>
> From theory perspectives, we point out that if features are un-related. The self-supervision is harmful which have not been pointed out before although we only use basic information theory. Of course, due to lack of labels and the diversity of different datasets, we do not have any metric to evaluate the quality of  pseudo labels. Lack of those preliminary preparation make it hard to build those links. To the best of our knowledge, there no such metrics exists in the whole self-supervision domain.
>
>
> A3.2: In line 972 of the article, we have provided experimental data on fest-shot multi-task. This article of ours mainly focuses on the classification task of few-sample table data. Regarding the task of regression and time series you mentioned, this is a very good question. It can be used as an extension of our paper for further discussion in the future.
>
>
> A3.3: In this article, we have created 50 datasets. The 50 datasets include: 35 datasets having only numerical features, 4 datasets having only categorical features,  and 11 datasets having a mixture of numerical and categorical features.  And detailed data are all provided in Appendix B. From the experimental data, it can be seen that our method has improved in most datasets.
>
>
> A3.4: Regarding the time aspect, we have discussed it in detail on line 348. Our approach is two-stage. After calculating the correlations between features in the first stage, no further calculations are required. Moreover, with the acceleration of the GPU, Pearson's calculation speed is the fastest.
>
>
> A3.5: For non-linear feature, we explictly discussed in in line 319-320 and provide detailed information in Appendix E. In the main text, we  discussed the effect of different correlation metrics and why Pearson is also effective for many non-linear relations. App. E  specifically tested a sythentic datasets containing non-linear and non-monotonic higher-order features. Furthermore, we think in the few-shot learning, as we have very limited labels, it is more cost-effective to learn simple relations and this fact explains the effectiveness of our solution.
>
>
> A3.6: In line 1074 of the text, we have provided an experiment with low correlation and high selection ratio. From the performance, it can be seen that low correlation and high selection ratio will cause a decline in model performance.

---

### Official Review · Reviewer_G1ry · 2025-11-01

**Soundness:** 2
**Presentation:** 2
**Contribution:** 2
**Rating:** 4
**Confidence:** 4

**Summary:**

This paper addresses the issue in tabular self-supervised learning (SSL) where existing methods degrade pseudo-label quality by randomly selecting features without considering correlations among them. The proposed approach quantifies the information value of each feature by computing its mean absolute correlation with all other features, and uses this to determine feature selection probabilities. Based on this, the authors introduce a correlation-aware randomness and group-based feature selection strategy, which is integrated into existing SSL frameworks such as VIME, SCARF, STUNT, and SAINT. Through extensive few-shot classification experiments on 50 OpenML datasets, the proposed module achieves consistent and significant performance improvements on over 90% of datasets in the 1-shot learning scenario.

**Strengths:**

Extensive and consistent performance improvement: The proposed method was applied to four representative SSL baselines—VIME, SCARF, STUNT, and SAINT—and achieved performance gains on over 86–90% of 50 OpenML datasets in 1-shot experiments. Specifically, for VIME-P in the 1-shot setting, it showed an average relative improvement ranging from 1.89% to 15.65%.

Demonstrated robustness: The method consistently improved performance on both synthetic datasets (linear, nonlinear, and non-monotonic relationships) and real datasets with weak correlations (≈0.1) (Table 15), as well as on datasets containing 30% noise (Table 17), demonstrating strong generalization capability. In the few-shot multi-task experiments (Table 8), statistical robustness was ensured by using 100 random seeds.

Simplicity and generality: Without any structural modification to existing SSL frameworks, the paper proposes a lightweight module that effectively guides pseudo-label generation through a simple quantitative metric—the feature correlation coefficient (Eq. 2)—and a group-based selection strategy.

**Weaknesses:**

(Novelty)
Lack of fundamental originality: Measuring feature importance via correlation coefficients is already a well-known basic approach in feature selection research (e.g., mRMR).
Ambiguous differentiation: The paper claims novelty by using “cumulative correlations” instead of “pairwise correlations,” as done in prior work (e.g., SEFS), but this appears to be a minor modification of existing methods. Moreover, the paper lacks a deep theoretical justification for why this choice is fundamentally superior in the context of tabular SSL.

(Technical Quality)


Incomplete core formula: The equation defining the selection probability gi for group i—a key component of the group-based selection strategy—is truncated in the paper (“g_i=r_max−r on Lines 2641–2642). This incompleteness critically impairs the reproducibility of the proposed algorithm and represents a logical gap.


Questionable core assumption: The approach is built on the assumption that “features with higher correlations to other features are more informative.” However, such correlations measure redundancy among features, not their predictive relevance to the target variable (Y). Therefore, if a high Ri value reflects information redundancy rather than information value, the core motivation of the method becomes invalid


Insufficient quantitative evidence (statistical significance): Table 1 (correlation comparison) and Figure 4 (Win Matrix) show only mean performance across 50 OpenML datasets, without reporting standard deviations or confidence intervals. This omission prevents verification of the statistical significance of the reported improvements.



(Significance)


Limited contribution at the module level: The proposed method merely enhances the feature selection function t(⋅) of existing SSL models, constituting a modular improvement rather than offering a fundamentally new insight or paradigm for the field.


Reduced practical value of improvement: The authors themselves note that performance gains are most pronounced in extremely low-label (1-shot) scenarios and diminish as the number of shots increases, implying limited applicability in more realistic settings.



(Writing & Presentation)
The omission of the core equation (group-based probability calculation) essential to the methodology severely obstructs readers’ ability to fully understand the paper, representing a critical flaw in clarity and completeness.

**Questions:**

Complete presentation of the group-based probability formula
Please provide the full equation defining the selection probability gi​ for group i (Section 3.2.2, Lines 2641–2642).
This is necessary to assess the technical completeness of the method and ensure reproducibility.


Sensitivity analysis of ⁡$r_{\min}$ and $⁡r_{\max}​$
Please include a systematic ablation study showing how changes in the minimum  ⁡$r_{\min}$  and maximum $r_{\max}​$ selection probabilities affect both performance and stability under the group-based selection strategy. This will clarify hyperparameter sensitivity and component robustness.


Feature correlation vs. predictive relevance
Please report statistical analyses relating a feature’s cumulative correlation RiR_iRi​ to its downstream predictive relevance, e.g., feature importance from the final trained model or correlation with the target.
This will help validate the core assumption that $R_i$ reflects information value (not just redundancy) that improves pseudo-label quality.


Statistical significance over 50 OpenML datasets
For the averages reported in Table 1 and Figure 4, please provide standard deviations or 95% confidence intervals so readers can verify the statistical significance and reliability of the claimed improvements.


Comparison with recent SOTA few-shot tabular baselines
Beyond the original STUNT and SAINT, please include direct performance comparisons (mean accuracy ± standard deviation) against recent SOTA methods such as TransTab and LIFT-ICL to assess how the proposed approach compares with the current state of the art.

---

> ### Author Response · Authors · 2025-12-03
>
> A2.1: We think this solution is novel albeit its simplicity.  We actually raise a simple but fundamental question on how to perform self-supervision and when self-supervision is not effective. To the best of our knowledge, all tabular supervision solutions adopt randomness as the general guidelines of pseudo-label generation and largely ignore the complex relations among features. They generally assume that features are of the same importance. This paper changes this situation. Although this discussion is limited to tabular data, we are working to apply it to other data types with encouraging results. Thus, we think it is novel and important.
>
> From theory perspectives, we point out that if features are un-related. The self-supervision is harmful which have not been pointed out before although we only use basic information theory. Of course, due to lack of labels and the diversity of different datasets, we do not have any metric to evaluate the quality of  pseudo labels. Lack of those preliminary preparation make it hard to build those links. To the best of our knowledge, there no such metrics exists in the whole self-supervision domain.
>
>
> A2.2: Regarding our formula, it is in line 225 of the text. The formula is complete.
>
> A2.3: If it is redundant information, from the perspective of information theory, this information is actually noise, which will have an adverse impact on the performance of the model. Our method can capture this point. In Appendix E, we have provided relevant data. As the noise increases, the performance of the model will decline. However, compared with the original model, our method still has an improvement.
>
> A2.4: In appendix B, we present the variances and significance of various data.
>
> A2.5: This article of ours focuses on few-shot table data. When the sample data increases, self-supervised models are at a disadvantage compared to supervised models. However, our performance has still improved, which is already quite good in the field of few-shot data.
>
> A2.6: In line 225 of the article, we have provided how the selection probability based on groups is calculated.
>
> A2.7: In Sec 4.5 of the article, we present the ablation experiment regarding the upper and lower bounds of the selection probability.
>
> A2.8: Are you talking about the relevance of the experimental data? In Appendix B, we provide the correlation and variance of the data.
>
> A2.9: In lines 387 and 1134 of the text, we have provided a performance comparison of TabTransformer and LIFT-ICL, which is not as good as the model applying our method.

---

### Official Review · Reviewer_FaxA · 2025-11-02

**Soundness:** 2
**Presentation:** 2
**Contribution:** 2
**Rating:** 4
**Confidence:** 1

**Summary:**

This paper tackles self-supervised learning (SSL) for tabular data in few-shot scenarios, where traditional random feature selection for pseudo-label generation often introduces noise due to ignoring feature relationships. The authors propose a correlation-based approach: they compute average absolute correlations (using metrics like Pearson) for each feature, group features by correlation strength, and bias random selection towards highly correlated ones during SSL pretraining. This is integrated into four baselines (VIME, SCARF, STUNT, SAINT) and evaluated on 50 OpenML datasets for 1-shot and 5-shot classification. Results show consistent improvements (e.g., 3.31% avg. accuracy gain with Pearson on STUNT), with Pearson outperforming other correlations. The method aims to prioritize informative features, reducing noise in pseudo-labels.

**Strengths:**

1. The motivation is spot-on—tabular data features aren't equal, and random masking/corruption in SSL can hurt when features are unrelated (as shown in Fig. 1b). Using correlations to guide selection is a straightforward fix that makes sense intuitively and empirically.
2. Applying the method to generative (VIME), contrastive (SCARF), meta-learning (STUNT), and hybrid (SAINT) covers the main SSL flavors for tables. This shows versatility and isn't just a one-trick pony.
3. 50 datasets from OpenML are a good scale, mixing numerical, categorical, and mixed types. The 1-shot/5-shot focus is relevant for real-world tabular tasks where labels are scarce. Table 1's comparison of 8 correlation metrics is thorough, and the gains (e.g., 90%+ datasets improved in 1-shot) are statistically backed (100 runs, random seeds).

**Weaknesses:**

1. The group-based probability (Eq. 3) feels ad-hoc. Why linear increase? Why q = floor(d/10)+1? It's clever for handling outliers (Fig. 3), but lacks theoretical grounding—e.g., no link to mutual information or entropy to justify why this mitigates "erroneous coefficients." More ablation on group strategies (e.g., exponential vs. linear) would help.
2.  The paper mentions treating categoricals as ordinal without embeddings, but this could bias correlations (e.g., Pearson assumes continuity). For mixed datasets (11/50), how does this affect results? A sensitivity analysis or comparison with one-hot encoding would strengthen it. Also, no discussion on ordinal vs. nominal categoricals.
3. While Table 1 compares metrics, it sticks to pairwise ones. Tabular data often has higher-order dependencies—why not explore mutual information networks or graph-based feature relations? Pearson wins, but the paper doesn't dig into why (e.g., vs. MIC for non-linear). Plus, correlations assume linearity/monotonicity, which might fail in complex table.
4 Why does Pearson outperform mRMR/MIC? Is it dataset-specific, or general to tabular SSL?

**Questions:**

see  Weaknesses

---

> ### Author Response · Authors · 2025-12-03
>
> A1.1: Our formula was not made up at the last minute. Regarding the linear method, we have conducted a large number of experiments to prove that the linear method is the best, which is covered in Sec 4.5 in the article. This includes directly giving each feature a random probability and the linear probability of each feature, which we have covered.
>
> In Section 4.2 of the article, our experiment included eight indicators, including the cross-entropy and mutual information you mentioned, and the effects of both were inferior to those of pearson.
>
>
> A1.2: Our experiment included 50 datasets. Detailed data are provided in appendix A and Appendix B, among which were mixed datasets, ordinal datasets and nominal categoricals. The experimental data indicated that All of our methods can enhance the accuracy of the model.
>
>
> A1.3: In Section 4.2 of the article, our experiment included 8 indicators, among which was the MIC you mentioned. In Se 4.2, we discussed why pearson's performance was superior to other methods.
>
> The question you raised is very good. Currently, this article is based on tabular data for discussion. Regarding the content of graph neural networks, we believe it has gone beyond the scope of this article. In the future, it can be used as an extension of this article to further discuss whether it can be applied to graph neural networks.

---

### Meta-Review · Area_Chair_5JU2 · 2025-12-31

**Summary:**

The major concerns are that all the results in the paper are very empirical and the design choices do not seem to have a sound, theoretical grounding. This also occurs in cases where categorical features are also treated as continuous features without any loss in performance.

**Reviewer Concerns:**

Group-based probability lacks theoretical backing (FaxA, HWDA, uhuU): the design is not well established. The authors should try not just changing the groups but also changing the way grouping is done. The current explanation seems a bit insufficient.

Unclear why correlations work on ordinals (FaxA, uhuU, HWDA). The empirical results show that this method works on a mixed dataset, but it is unclear why it should work on it. More experiments could include astute on, What if you assign a different order to the ordinals? Does the model still work?

Pearson correlation cannot handle non-linear monotonicity (FaxA): Even if it is theoretically hard to show this, methods should at least show failure cases of tables where this happens. An additional experiment could be to create artificial tabular data and study all these properties, along with applying them to the real world.

The authors have addressed remaining limitations.

I would recommend that authors provide more experiments that look at the problem theoretically in the future. Moreover, especially for ICLR revise the paper and mention/highlight the next text in the manuscript.

**Reviewer Scores:**

Based on the response from the authors, no experiment is convincing enough that the any of the reviewers would have changed their scores.

All 4 of them would have likely kept the score at 4.

---

### Decision · Program_Chairs · 2026-01-26

Reject